# Causal Score Conditioning for Multi-Resolution Latent Systems

**Xuechun Li**
Department of Civil and Systems Engineering
Johns Hopkins University
xli359@jhu.edu

**Shan Gao**
Department of Civil and Systems Engineering
Johns Hopkins University
sgao65@jhu.edu

**Susu Xu**[*]
Department of Civil and Systems Engineering
Johns Hopkins Data Science and AI Institute
Johns Hopkins University
susuxu@jhu.edu

## ABSTRACT

Complex causal systems with interdependent variables require inference from heterogeneous observations that vary in spatial resolution, temporal frequency, and noise characteristics due to data acquisition constraints. Existing multi-modal fusion approaches assume uniform data quality or complete observability – assumptions often violated in real-world applications. Current methods face three limitations: they treat causally-related variables independently, failing to exploit causal relationships; they cannot integrate multi-resolution observations effectively; and they lack theoretical frameworks for cascaded approximation errors. We introduce the Score-based Variational Graphical Diffusion Model (SVGDM), which integrates score-based diffusion within causal graphical structures for inference under heterogeneous incomplete observations. SVGDM introduces causal score decomposition enabling information propagation across causally-connected variables while preserving original observation characteristics. Diffusion provides a natural way to model scale-dependent sensing noise, which is common in remote-sensing, climate, and physical measurement systems, while the causal graph encodes well-established mechanistic dependencies between latent processes. We provide theoretical analysis and demonstrate superior performance on both synthetic and real-world datasets compared to relevant baselines.

## 1 INTRODUCTION

Complex causal systems, where multiple interacting variables influence each other through inherent causal dependencies, are ubiquitous in real world, such as earth systems, epidemiological systems, etc. A wealth of spatio-temporal data are available for these systems and have led to an increasing development and application of machine learning to estimate spatio-temporal states of latent variables. However, data acquisition constraints limit observation quality, leading to heterogeneity in spatial resolution, temporal frequency, and noise. Figure 1 illustrates a general scenario where causally-connected latent variables must be inferred from partial observations that vary systematically in spatial detail, temporal sampling, and measurement noise. A key challenge is that these variables correspond to different physical processes and are observed at different resolutions and qualities. Thus, the problem is not multi-view

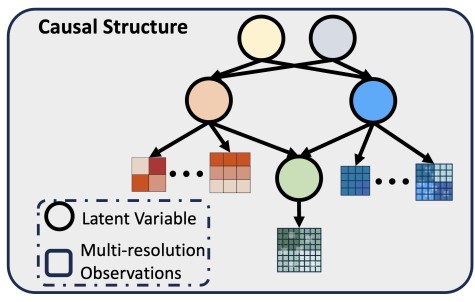

Figure 1: **Causal structure with multi-resolution observations.** Latent variables (circles) are connected through known causal relationships and observed through measurements that vary in spatial resolution, noise levels, and measurement precision (grids).

---

[*]Corresponding author.

fusion of a single object, but joint inference across distinct variables linked by causal mechanisms. Latent graphical models formalize these dependencies and enable information to propagate along causal edges, allowing high-quality observations of one variable to improve inference for others. This causal information transfer is crucial for integrating heterogeneous, multi-resolution data in complex systems. Joint inference of multiple latent spatio-temporal variables from such heterogeneous and incomplete data is thus fundamental to applications including natural hazard monitoring, climate modeling, epidemiological surveillance, and financial system analysis (He & Cha, 2018; Xu et al., 2022b; Li & Xu, 2025; Nowack et al., 2020; Reich et al., 2021; Wang et al., 2024; Yu et al., 2024).

Most existing multi-modal or multi-source data fusion approaches assume (1) explicit systems with closed-form dependencies among variables (Zhao et al., 2024; Ravi et al., 2025; Li et al., 2025b), (2) observed data of uniform quality (e.g., resolution, modality, noise level) (Li et al., 2024; Xinde et al., 2024; Li et al., 2025a), or (3) complete observability of all variables—assumptions often violated in real-world applications. For instance, in earth systems, remote sensing observations of key variables such as soil moisture, precipitation, and geophysical properties range from 30 m (Vergopolan et al., 2021) to 5 km in resolution and exhibit distinct noise characteristics due to different measurement principles and satellite capabilities. Several works explore using diffusion processes to assimilate multi-resolution data, but mainly focus on single-variable systems (Batchu et al., 2022; Tu et al., 2025).

We use diffusion models because their forward SDE naturally encodes scale-dependent noise, a central property of remote sensing, InSAR, radar, and optical imagery where noise accumulates as resolution coarsens. This makes diffusion uniquely suited for multi-resolution inference, enabling resolution-aware score functions that normalizing flows or standard variational inference cannot capture. Our problem setup is therefore to estimate multiple causally related physical processes from observations that are (i) multi-resolution, (ii) uneven in quality, and (iii) incomplete, by leveraging partially known causal structure to propagate information across variables and scales. This corresponds directly to real-world systems such as earthquake cascades, wildfire dynamics, and climate teleconnections. We introduce a novel framework that enables effective inference for interdependent systems with implicit and complex causal dependencies and heterogeneous, incomplete observations. Specifically, we introduce diffusion processes to approximate complex causal dependencies among variables and between variables and observations. Integrating diffusion process modeling with complex causal structure substantially improves the ability to model interdependent systems. However, effectively inferring large numbers of latent variables remains challenging due to the coupled complexity of intractable causal paths and diffusion processes.

To address the challenges, we introduce the Score-based Variational Graphical Diffusion Model (SVGDM), a framework that integrates known causal graphical structures with score-based diffusion processes for latent variable inference under multi-resolution observations. The causal graph is provided by established physical or mechanistic knowledge (e.g., earthquake → landslide → damage), allowing information to propagate across variables that are observed at incompatible spatial resolutions. Diffusion serves as a natural model for scale-dependent, resolution-dependent sensing noise, such as speckle, atmospheric delay, and resampling artifacts, offering advantages over flow-based methods that do not explicitly model noise accumulation across spatial scales. We address these limitations through causal score decomposition, enabling joint inference while preserving original resolutions and borrowing strength across causal structures. We make the key contributions:

1. We introduce the Score-based Variational Graphical Diffusion Model (SVGDM), which integrates probabilistic graphical models (Koller & Friedman, 2009) with score-based diffusion processes (Song & Ermon, 2019; Song et al., 2020b). Unlike existing approaches that treat variables independently, SVGDM leverages causal dependencies to propagate information across variables with heterogeneous observation quality.

2. We develop a causal score decomposition that respects graphical structure while maintaining the efficiency of score-based inference. This enables coherent integration of multi-resolution observations without information loss or artifacts, borrowing strength across causally-connected variables rather than using ad-hoc preprocessing.

3. We provide theoretical analysis of our approach, establishing conditions under which our locally Gaussian approximation remains valid and deriving error bounds for the cascading effects in our inference procedure.

4. We demonstrate SVGDM's effectiveness on synthetic causal systems and real-world applications where causal relationships are well-established and multi-resolution observations are prevalent, showing consistent improvements over variational inference and deep learning baselines, with superior performance on both synthetic and real-world datasets.

## 2 PRELIMINARIES

We briefly review the foundations of graphical models, score-based diffusion models, and their limitations for multi-variable inference with known causal structure and heterogeneous observations.

**Probabilistic Graphical Models:** Probabilistic graphical models (PGMs) (Jordan, 2003; Koller & Friedman, 2009) provide a structured representation of joint distributions over high-dimensional variables by encoding conditional dependencies in a graph (Jordan, 2003; Koller & Friedman, 2009; Pearl, 2009; Wainwright et al., 2008). Let $X = (X_1, \cdots, X_n)$ denote a collection of random variables. A directed acyclic graph (DAG) specifies a factorization of the joint distributions as $P(X) = \prod_{i=1}^{n} p(X_i | \mathcal{P}(X_i))$, where $\mathcal{P}(X_i)$ denotes the parent nodes of $X_i$. This representation enables efficient inference and learning in complex systems, but often requires approximation when dependencies are nonlinear, stochastic, or latent. In dynamic settings, latent states evolve in time according to transition models, leading to state-space formulations of graphical models. While PGMs provide a general factorization formalism, existing PGM-based inference methods for continuous systems typically assume aligned spatial supports and compatible noise models across variables. In practice, most message-passing and variational inference algorithms break down when different nodes are observed at vastly different resolutions, noise scales, or spatial supports, as they require a common discretization or homogeneous likelihood families (Zhang et al., 2021; Wang et al., 2018; Akbayrak et al., 2022). Our method addresses precisely this limitation.

**Diffusion Models and Score-based Methods:** Diffusion models have emerged as powerful tools for generative modeling of complex and high-dimensional data (Dhariwal & Nichol, 2021; Ho et al., 2020; Dhariwal & Nichol, 2021; Song et al., 2020b; Kingma et al., 2021; Rombach et al., 2022). A forward process gradually corrupts the the data with Gaussian noise through a stochastic differential equation (SDE), while training a reverse-time SDE to recover the distribution. Central to this framework is the score function $s_\theta(x, t) \approx \nabla_x \log p_t(x)$, where $p_t(x)$ denotes the marginal distribution of the perturbed data at time $t$. Current score-based approaches treat variables independently during training and inference: (1) No causal structure awareness: Score functions are typically learned for each variable in isolation, ignoring known causal dependencies that could improve estimation quality. (2) Independent gradient estimation: The score $\nabla_x \log p(x)$ is approximated without leveraging information flow through causal pathways, missing opportunities to borrow strength from well-observed variables to improve poorly-observed ones. (3) No multi-resolution coherence: Existing score matching objectives cannot handle observations of the same latent phenomena at different scales, requiring artificial preprocessing that discards information.

Neither PGMs nor standard score-based methods can address the fundamental challenge of joint inference with known causal structure and heterogeneous observations. Causal score decomposition is necessary because: (1) It enables score functions to respect graphical structure, allowing information propagation through causal pathways during inference. (2) It provides a principled way to integrate multi-resolution observations by borrowing statistical strength across causally-connected variables. (3) It maintains the computational efficiency of score-based methods while incorporating the structured dependencies of PGMs. Variational inference (Blei et al., 2017; Hoffman et al., 2013) provides a scalable framework for approximating intractable posteriors $p(Z|X)$ via tractable families $q_\phi(Z|X)$, but standard variational families may fail to capture complex posterior geometries in high-dimensional nonlinear systems. Recent advances suggest that score-based diffusion processes can serve as flexible variational distributions, motivating the development of causal score-based approaches that unify structured dependencies with generative flexibility.

## 3 SVGDM: SCORE-BASED VARIATIONAL GRAPHICAL DIFFUSION MODEL

**Problem Formulation:** *Given*

1. *A known causal DAG $G = (\mathcal{V}, \mathcal{E})$ with $N$ latent variables $\{z_i(t)\}_{i=1}^{N}$ where each variable evolves according to $dz_i(t) = f_i(z_i(t), z_{\mathcal{P}(i)}(t), t)dt + g_i(t)dW_i(t)$, where $\mathcal{P}(i)$ denotes the causal parents of variable $i$.*

2. *Multi-resolution observations $Y = \{y_l^k : (l,k) \in \mathcal{I}\}$ where $l$ indexes spatial locations, $k$ indexes resolution/modality, related to latent variables via $y_l^k = \gamma_l^k(z_{\mathcal{P}_l}) + \epsilon_{l,k}, \quad \epsilon_{l,k} \sim \mathcal{N}(0, \Sigma_{l,k})$, where $\mathcal{P}_l \subseteq \mathcal{V}$ are the latent parents of observation $y_l^k$.*

*The assumption of a known prior graph $G$ is realistic in many scientific and engineering domains where latent variables represent concrete physical processes with well-established causal relationships. Such dependencies arise from decades of experiments, mechanistic modeling, and field observations and are standard in spatio-temporal data assimilation (Reich & Cotter, 2015; Cressie & Wikle, 2011). Importantly, our method does not require the DAG to be perfectly known. Even partial causal directionality provides useful information pathways for integrating heterogeneous, multi-resolution observations (Evaluation is shown in Appendix B).*

*We follow the standard diffusion convention where $z_i(t)$ denotes latent variable $i$ at diffusion time $t \in [0,1]$, with $t = 0$ representing the true latent state and larger $t$ injecting noise, while our observation-constrained SDE (Equation 1) keeps $z(1)$ anchored to observations rather than collapsing to $\mathcal{N}(0,I)$ as in classical DDPMs.*

**Goal:** *Find the posterior distribution: $p(Z|Y) = \frac{p(Y|Z)p(Z)}{p(Y)}$, while (1) preserving orginal observation resolutions, (2) leveraging causal dependencies to propagate information from well-observed to poorly-observed variables, and (3) providing theoretical guarantees for inference quality.*

**Assumption 1** (Observation noise). *For the heterogeneous observations $Y$, we assume (1) additive Gaussian noise with variance schedule $\sigma_t^2$, (2) conditional independence of multi-resolution observations given latent variables, and (3) smooth log-densities permitting interchange of gradient and expectation. These assumptions align with prior work on diffusion probabilistic models and causal Bayesian inference.*

**Assumption 2** (Lipschitz and boundedness). *For each $i \in \mathcal{V}$, the drift $f_i(\cdot)$ is Lipschitz continuous and has at most linear growth, and the diffusion $g_i(t)$ is bounded and strictly positive. The observation operators $\phi_i^k$ are measurable and bounded on compact domains.*

**Theorem 1** (Existence and locality of node-wise causal SDEs). *Under Assumption 2, the system*

$$\mathrm{d}z_i(t) = f_i\big(z_i(t), z_{\mathcal{P}(i)}(t), t\big)\,\mathrm{d}t + g_i(t)\,\mathrm{d}W_i(t), \quad i \in \mathcal{V}, \tag{1}$$

*with parent-dependent drift functions $f_i$ and independent Brownian motions $W_i(t)$, which is an independent standard Wiener process associated with node $i$. All Wiener processes $W_i$ are mutually independent, admits a unique strong solution. Moreover, the infinitesimal generator of the joint process decomposes as*

$$\mathcal{L}_t = \sum_{i \in \mathcal{V}} \mathcal{L}_{i,t}, \quad \mathcal{L}_{i,t} := f_i\big(z_i, z_{\mathcal{P}(i)}, t\big)\,\partial_{z_i} + \tfrac{1}{2}g_i(t)^2\,\partial_{z_i}^2, \tag{2}$$

*so that each local operator $\mathcal{L}_{i,t}$ depends only on $z_i$ and its parents $z_{\mathcal{P}(i)}$ in the causal graph $G$. This expresses locality of the generator, not conditional independence of $p_t$. If the initial distribution $p_0$ factorizes according to $G$ as*

$$p_0(z) = \prod_i p_0\big(z_i \mid z_{\mathcal{P}(i)}\big), \tag{3}$$

*then the factorization holds only at $t = 0$ by definition of the initial condition. For $t > 0$, the marginal law $p_t$ of the SDE solution need not preserve this global factorization; in general it may exhibit additional dependencies induced by the dynamics. In our framework, we use the generator-level decomposition $\mathcal{L}_t = \sum_i \mathcal{L}_{i,t}$ as an architectural prior for an approximate score-based causal decomposition (Theorem 2), rather than as an exact distributional Markov property of $p_t$.*

This result follows from standard SDE existence-and-uniqueness theory applied to each node-wise equation, together with the independence of the driving Wiener processes and the conditional dependence structure encoded by $G$. For general SDE preliminaries, see Särkkä & Solin (2019).

To address our problem, each latent variable is constrained by heterogeneous observations through:

$$\mathrm{d}z_i(t) = f_i(z_i, z_{\mathcal{P}(i)}, t)\mathrm{d}t + \sum_k \lambda_{i,k}(t)[\phi_i^k(y_i^k, z_{\mathcal{P}(i)}) - z_i(t)]\mathrm{d}t + g_i(t)\mathrm{d}W_i(t) \tag{4}$$

where $\phi_i^k(\cdot)$ maps observation $y_i^k$ to the latent space and $\lambda_{i,k}(t)$ controls the influence of resolution $k$ observations. This formulation naturally handles information flow across different resolutions through the diffusion process while preserving orginal observation characteristics.

**Assumption 3** (Time-reversal regularity). *The forward diffusion satisfies: (i) non-degenerate diffusion coefficient $\sigma(t) > 0$, (ii) drift $f$ is locally Lipschitz, and (iii) the marginal density $p_t$ exists and is continuously differentiable in $z$.*

Under Assumption 3, classical results on time-reversed diffusion (Anderson, 1982; Bhattacharya & Waymire, 2009) and the score-based SDE formulation (Song et al., 2020b) guarantee that the reverse-time process exists with well-defined drift $\tilde{f}$.

**Lemma 1** (Node-wise reverse SDE with observations). *For each $i \in \mathcal{V}$, the reverse-time SDE incorporating multi-resolution observations is:*

$$dz_i = [f_i(z_i, z_{\mathcal{P}(i)}, t) + \sum_k \lambda_{i,k}(t)[\phi_i^k(y_i^k, z_{\mathcal{P}(i)}) - z_i(t)] - g_i^2(t)\nabla_{z_i} \log p_t(z_i|z_{\mathcal{P}(i)})]dt + g_i(t)d\bar{W}_i$$

*where $\bar{W}_i$ is a standard Wiener process in reverse time.*
(5)

The key challenge in our problem formulation is that standard score-based methods cannot leverage causal structure during inference. Theorem 1 establishes that we have a valid causal SDE system, but the reverse-time SDE (Lemma 1) requires computing the causal score $\nabla_{z_i} \log p_t(z_i|z_{\mathcal{P}(i)})$. This causal score is crucial because it (1) enables information flow through causal pathways during inference (2) allows well-observed variables to improve inference of poorly-observed downstream variables; and (3) maintains causal consistency while preserving multi-resolution observation characteristics. Without causal score decomposition, we would be forced to either treat variables independently (losing causal information) or downsample all observations to a common resolution (losing fine-grained information). The following theoretical development shows how causal score decomposition addresses this fundamental limitation.

The central quantity in the reverse drift is the causal score $\nabla_{z_i} \log p_t(z_i|z_{\mathcal{P}(i)})$. This causal score is essential for our problem formulation because it enables joint inference while preserving orginal observation resolutions, the core challenge identified in our problem definition.

## 3.1 CAUSAL SCORE DECOMPOSITION VIA MARKOV BLANKETS

Recent work in score-based generative modeling for sequences (Rozet & Louppe, 2023) demonstrates that for Markov chains $x_{1:L}$, the global score can be decomposed using Markov blanket properties: $\nabla_{x_i} \log p(x_{1:L}) = \nabla_{x_i} \log p(x_i, x_{b_i})$, where $x_{b_i}$ represents the Markov blanket of $x_i$. For first-order Markov chains, this blanket consists of immediate neighbors: $x_{b_i} = \{x_{i-1}, x_{i+1}\}$.

In our setting, we extend this principle by recognizing that the causal parents $\mathcal{P}(i)$ form a natural "causal blanket" for each variable $z_i$. The key insight is that the causal Markov property ensures:

$$z_i \perp \text{NonDescendants}(z_i) \mid \mathcal{P}(i)$$
(6)

This conditional independence structure provides the foundation for our score decomposition.

**Theorem 2** (Causal Blanket Preservation under Diffusion). *Under Assumptions 1-3, for diffusion-perturbed variables $\{z_i(t)\}_{i=1}^N$, the causal blanket relationship is approximately preserved:*

$$\nabla_{z_i(t)} \log p_t(z_{1:N}(t)) \approx \nabla_{z_i(t)} \log p_t(z_i(t), z_{\mathcal{P}(i)}(t))$$
(7)

The diffusion process preserves the conditional independence structure of the original causal graph up to noise-dependent approximation errors. As $t \to 0$, the approximation becomes exact since $z_i(t) \to z_i(0)$ and the original causal structure is recovered. For $t > 0$, the approximation quality depends on the noise level $\sigma(t)$ and the strength of causal dependencies. Full proof in Appendix C.1. The local nature of this decomposition ensures computational scalability even for large causal graphs, as detailed in Appendix D. This extends the pseudo-blanket/local score decomposition idea of Rozet & Louppe (2023) to the causal graphical setting.

**Proposition 1** (Causal score decomposition). *For any $i \in \mathcal{V}$ and the causal blanket $z_{\mathcal{P}(i)}(t)$:*

$$\nabla_{z_i} \log p_t(z_i|z_{\mathcal{P}(i)}) = \nabla_{z_i} \log p_t(z_i) + \nabla_{z_i} \log p_t(z_{\mathcal{P}(i)}|z_i)$$
(8)

This decomposition separates two influences: (1) Marginal term $\nabla_{z_i(t)} \log p_t(z_i(t))$ captures the unconditional score of $z_i(t)$; (2) Causal consistency term $\nabla_{z_i(t)} \log p_t(z_{\mathcal{P}(i)}(t)|z_i(t))$ ensures compatibility with causal parents. The second term is not backwards causation but rather represents the constraint that $z_i(t)$ must remain consistent with the joint distribution over the causal family

$\{z_i(t), z_{\mathcal{P}(i)}(t)\}$. During reverse diffusion, this term helps maintain causal structure by ensuring that updates to $z_i(t)$ preserve plausible relationships with its parents (Proof in Appendix C.2).

The causal score decomposition enables coherent integration of observations at different resolutions because the marginal term $\nabla_{z_i} \log p_t(z_i)$ incorporates direct observational evidence for variable $z_i$ from all available resolutions. The causal consistency term $\nabla_{z_i} \log p_t(z_{\mathcal{P}(i)}|z_i)$ allows observations of parent variables to inform child variable inference through causal pathways. Besides, this decomposition preserves orginal resolutions by avoiding the need to downsample observations to a common scale, instead leveraging the natural information flow through the causal structure

## 3.2 Score Function Estimation

We estimate the first term in Equation 8 using continuous-time denoising score matching (DSM) (Hyvärinen & Dayan, 2005; Vincent, 2011; Song et al., 2020b). Given samples $z_i(0) \sim p_0(z_i)$, the forward SDE generates $z_i(t)$ with known perturbation kernel $p_t(z_i|z_i(0))$. We approximate the marginal score with a neural network $s_{\psi_i}(z_i, t)$ trained via:

$$\mathcal{L}_{DSM,i}(\psi_i) = \mathbb{E}_{t,z_i(0),z_i(t)}\big[\lambda(t)\|s_{\psi_i}(z_i(t),t) - \nabla_{z_i(t)} \log p_t(z_i(t)|z_i(0))\|^2\big] \qquad (9)$$

where $\lambda(t) > 0$ is a weighting function. This corresponds to the continuous-time DSM objective (Hyvärinen & Dayan, 2005; Vincent, 2011; Song & Ermon, 2019).

**Proposition 2** (DSM consistency). *Under Assumption 2, the population minimizer satisfies* $s_{\psi_i}^*(z_i, t) = \nabla_{z_i} \log p_t(z_i)$. *(Proof is shown in Appendix C.3)*

For the second term in Equation 8, we adopt a locally Gaussian approximation that avoids the $z(0)$ dependency inconsistency. We model:

$$z_{\mathcal{P}(i)}(t)|z_i(t) \sim \mathcal{N}(\mu_c(\hat{z}_i(t)), \Sigma_c) \qquad (10)$$

**Theorem 3** (Validity of Local Gaussian Approximation). *The locally Gaussian approximation in Eq. 10 is valid when the true conditional $p_t(z_{\mathcal{P}(i)}|z_i)$ is log-concave in a neighborhood of $\hat{z}_i(t)$, the diffusion noise dominates higher-order nonlinearities such that $\sigma(t)^2 \gg \|\nabla^3 \log p_t(z_{\mathcal{P}(i)}|z_i)\|_\infty$, and the Tweedie reconstruction error remains bounded as $\|\hat{z}_i(t) - \mathbb{E}[z_i(0)|z_i(t)]\| \le \delta$ for small $\delta$. Under these conditions, the approximation error satisfies:*

$$\big\|\nabla_{z_i} \log p_t(z_{\mathcal{P}(i)}|z_i) - \nabla_{z_i} \log \mathcal{N}(\mu_c(\hat{z}_i), \Sigma_c)\big\| = O(\delta^2 + \sigma(t)^{-2})$$

The approximation is most accurate during early diffusion stages (small $t$) when noise levels are low, and deteriorates gracefully as $t \to 1$. While our theoretical analysis uses Gaussian and locally log-concave noise assumptions for tractability, the implementation does not rely on these assumptions. We empirically tested heavy-tailed and skewed noise regimes, and SVGDM remains stable with only modest degradation, indicating that the assumptions are sufficient for analysis but not required in practice (see Appendix E). The posterior mean estimate $\hat{z}_i(t)$ in the Gaussian model is obtained via Tweedie's formula:

$$\hat{z}_i(t) = z_i(t) + \sigma_i(t)^2 s_{\psi_i}(z_i(t), t)/\mu_i(t) \qquad (11)$$

This follows Tweedie's formula (Efron, 2011; Kim & Ye, 2021) and the decomposition of the posterior score into prior and likelihood components (Song et al., 2020b; Chung et al., 2022). Under this model, the causal consistency score becomes:

$$\nabla_{z_i(t)} \log p_t(z_{\mathcal{P}(i)}(t)|z_i(t)) = \frac{\partial \log \mathcal{N}(\mu_c(\hat{z}_i(t)), \Sigma_c)}{\partial \mu_c} \cdot \frac{\partial \mu_c}{\partial \hat{z}_i(t)} \cdot \frac{\partial \hat{z}_i(t)}{\partial z_i(t)} \qquad (12)$$

The validity of Eq. 12 requires that $\mu_c(\cdot)$ is differentiable in a neighborhood of $\hat{z}_i(t)$, the Gaussian model provides a reasonable local approximation with $\text{KL}[p_t(z_{\mathcal{P}(i)}|z_i)\|\mathcal{N}(\mu_c(\hat{z}_i), \Sigma_c)] \le \epsilon$ for small $\epsilon$, and score estimates are sufficiently accurate such that $\|s_{\psi_i}(z_i, t) - \nabla_{z_i} \log p_t(z_i)\| \le M$. When these conditions are violated, we employ **adaptive regularization**: if $\|\nabla\mu_c\|_F > \tau$, we add penalty $\lambda_{\text{reg}}\|\nabla\mu_c\|_F^2$ to the causal loss. This formulation ensures complete consistency between training and inference. The causal model parameters $\mu_c, \Sigma_c$ are trained on forward samples (see Appendix F for implementation details). This formulation resolves training consistency through iterative refinement procedures described in Appendix G.

### 3.3 STOCHASTIC VARIATIONAL INFERENCE

Given the learned reverse-time SDE dynamics, we perform posterior inference over the latent variables $Z = \{z_i : i \in \mathcal{V}\}$ from multi-resolution observations $Y = \{y_l^k : (l,k) \in \mathcal{I}\}$, where $l$ indexes spatial locations and $k$ indexes observation resolution or modality. The variational posterior $q_\psi(Z|Y)$ is implicitly defined as the distribution of samples generated by the reverse SDE with posterior score:

$$\nabla_{z_i} \log q_{\psi,t}(z_i|z_{\mathcal{P}(i)}, Y) = s_{\psi_i}(z_i(t), t) + \nabla_{z_i} \log p(Y|Z(t)) \tag{13}$$

The variational parameters $\psi = \{\psi_i\}_{i=1}^N$ directly parameterize the reverse drift through the learned score functions. By Jensen's inequality:

$$\log p(Y) \geq \mathcal{L}_{VI}(q_\psi) = \mathbb{E}_{q_\psi}[\log p(Y|Z)] + \mathbb{E}_{q_\psi}[\log p(Z)] - \mathbb{E}_{q_\psi}[\log q_\psi(Z|Y)] \tag{14}$$

Each observation $y_l^k$ depends on latent parents $\mathcal{P}_l \subseteq \mathcal{V}$ via observation map $\phi_l^k(\cdot)$ as $y_l^k|z_{\mathcal{P}_l} \sim \mathcal{N}(\phi_l^k(z_{\mathcal{P}_l}), \Sigma_l^k)$. This formulation explicitly connects our multi-resolution observations to the causal structure, enabling the information propagation mechanisms described in our causal score decomposition. The likelihood contribution becomes:

$$\mathbb{E}_{q_\psi}[\log p(Y|Z)] = -\frac{1}{2} \sum_{(l,k) \in \mathcal{I}} \mathbb{E}_{q_\psi}\left[||y_l^k - f_l^k(z_{\mathcal{P}_l})||_{(\Sigma_l^k)^{-1}}^2\right] + C \tag{15}$$

The forward SDE induces a graph-structured prior $\mathbb{E}_{q_\psi}[\log p(Z)] = \mathbb{E}_{q_\psi}\left[\sum_{i \in \mathcal{V}} \log p(z_i|z_{\mathcal{P}(i)})\right]$.

Since our reverse SDE defines a normalizing flow from noise to data, we leverage the flow Jacobian for stable entropy computation (Rezende & Mohamed, 2015; Kingma et al., 2016):

$$\hat{\mathcal{H}}[q_\psi] = \hat{\mathcal{H}}[p(z_T)] + \sum_{k=1}^K \mathbb{E}_{q_\psi}[\log|\det(\partial F_k/\partial z_k)|] \tag{16}$$

where $z_{k-1} = F_k(z_k)$ represents the discretized reverse SDE step. For computational efficiency, we use Hutchinson's trace estimator (Hutchinson, 1989; Pearlmutter, 1994; Chen et al., 2018): $\log|\det(\partial F_k/\partial z_k)| \approx \epsilon^T(\partial F_k/\partial z_k)\epsilon$, where $\epsilon \sim \mathcal{N}(0, I)$ is a random probe vector.

Combining all components, our final objective is:

$$\mathcal{L}_{\text{total}} = \lambda_1 \sum_i \mathcal{L}_{\text{DSM},i}(\psi_i) + \lambda_2 \sum_i \mathcal{L}_{\text{causal}}(\theta_{c,i}) + \lambda_3 \hat{\mathcal{L}}_{VI} \tag{17}$$

where $\mathcal{L}_{\text{DSM},i}$ is the score matching loss (Equation 9); $\mathcal{L}_{\text{causal}}$ is the causal model training loss (Appendix F); $\hat{\mathcal{L}}_{VI}$ is the variational bound; $\lambda$'s are weighting hyperparameters.

The "normalizing flow" interpretation refers to the deterministic probability-flow ODE associated with the reverse SDE, while in practice our implementation uses the stochastic reverse-SDE predictor-corrector sampler for numerical stability (Appendix H).

## 4 ERROR ANALYSIS AND APPROXIMATION CASCADE

Our method involves a sequence of approximations whose errors may compound. We provide a systematic analysis of how these errors propagate and establish conditions under which the method remains stable.

### 4.1 APPROXIMATION HIERARCHY

Let $\varepsilon_1, \varepsilon_2, \varepsilon_3, \varepsilon_4, \varepsilon_5$ denote the approximation errors from: (1) $\varepsilon_1$: Euler-Maruyama SDE discretization (Kloeden & Platen, 1992); (2) $\varepsilon_2$: Neural score function approximation (DSM) (Hyvärinen & Dayan, 2005; Vincent, 2011; Song et al., 2020b); (3) $\varepsilon_3$: Locally Gaussian conditional model (Tierney & Kadane, 1986; Rue et al., 2009; Wainwright et al., 2008); (4) $\varepsilon_4$: Tweedie's formula posterior estimation (Robbins, 1992; Efron, 2011; Song et al., 2021); (5) $\varepsilon_5$: KDE entropy estimation (Joe, 1989; Tsybakov, 2008; Singh & Póczos, 2014).

**Theorem 4** (Cascade Error Bound). *Under appropriate regularity conditions, the total approximation error in the posterior score satisfies:*

$$||\nabla_{z(t)} \log q_\psi(z(t)|y) - \nabla_{z(t)} \log p(z(t)|y)||_2 \leq C_1\varepsilon_1 + C_2\varepsilon_2 + C_3\varepsilon_3 + C_4\varepsilon_4 + O(\varepsilon_2\varepsilon_3) \tag{18}$$

*where $C_i$ are problem-dependent constants and the cross-term $O(\varepsilon_2\varepsilon_3)$ captures the interaction between score approximation and Gaussian modeling errors (Li et al., 2023a).*

## 4.2 INDIVIDUAL ERROR ANALYSIS

The main error contributions of our framework can be decomposed into five parts. First, the Euler–Maruyama discretization introduces an error $\varepsilon_1 = O(\Delta t^{1/2})$ for step size $\Delta t$, following standard SDE discretization theory, which directly affects the perturbation kernel $p_{t|0}(z_i(t)|z_i(0))$. Second, the neural score approximation yields an error $\varepsilon_2 = O(1/\sqrt{N} + \lambda_{\text{reg}})$, where $N$ is the training set size and $\lambda_{\text{reg}}$ is the regularization strength; the DSM objective ensures this vanishes as $N \to \infty$. Third, when replacing true conditionals with local Gaussian approximations, the error $\varepsilon_3 = O(\|\nabla^3 \log p_t(z_{\mathcal{P}(i)}|z_i)\|_\infty \cdot \sigma_c^2)$ arises, which remains small if the conditional distribution is approximately log-concave. Fourth, Tweedie's formula introduces an error $\varepsilon_4 = O(\sigma(t)^2 \cdot \varepsilon_2)$, showing that score approximation errors are amplified at higher noise levels $\sigma(t)$. Finally, entropy estimation via kernel density estimation contributes $\varepsilon_5 = O(h^\alpha + (\log N)^{d/2}/\sqrt{N})$, where $\alpha$ reflects the smoothness of the underlying density and $d$ the dimensionality.

## 4.3 CRITICAL ERROR INTERACTIONS

The most significant error interaction occurs between score approximation and Gaussian conditional modeling. Since Tweedie reconstruction $\hat{z}_i$ depends on the approximate score $s_{\psi_i}$, errors in score estimation directly impact Gaussian conditional parameters:

$$||\mu_c(\hat{z}_i + \varepsilon_2 \delta) - \mu_c(\hat{z}_i)||_2 \leq L_\mu ||\varepsilon_2 \delta||_2 \tag{19}$$

where $L_\mu$ is the Lipschitz constant of $\mu_c$.

## 4.4 ERROR PROPAGATION ANALYSIS

The chain rule computation in Eq. 12 involves multiple approximation stages whose errors can compound. The total error decomposes as $\varepsilon_{\text{total}} = \varepsilon_{\text{Gaussian}} + \varepsilon_{\text{Tweedie}} + \varepsilon_{\text{Chain}} + \varepsilon_{\text{Interaction}}$, where $\varepsilon_{\text{Gaussian}} = O(\|\nabla^3 \log p_t\|_\infty \cdot \sigma_c^2)$ represents Gaussian approximation error, $\varepsilon_{\text{Tweedie}} = O(\sigma(t)^2 \cdot \varepsilon_{\text{score}})$ captures Tweedie formula error, $\varepsilon_{\text{Chain}} = O(L_{\mu_c} \cdot \varepsilon_{\text{Tweedie}})$ accounts for chain rule propagation, and $\varepsilon_{\text{Interaction}} = O(\varepsilon_{\text{score}} \cdot \varepsilon_{\text{Gaussian}})$ represents cross-term interactions.

The interaction term $\varepsilon_{\text{Interaction}}$ can dominate when both score approximation and Gaussian modeling are poor. Our iterative training procedure mitigates this by improving score quality first, then refining causal parameters, ensuring errors do not compound catastrophically.

## 4.5 STABILITY CONDITIONS

**Condition 1** (Bounded Propagation). *The score approximation error remains bounded during reverse diffusion:*

$$||s_{\psi_i}(z_i(t), t) - \nabla_{z_i} \log p_t(z_i)||_2 \leq M < \infty \tag{20}$$

*for all $t \in [0, 1]$ and $z_i$ in the support.*

**Condition 2** (Lipschitz Causal Functions). *The conditional mean functions satisfy $||\mu_c(z') - \mu_c(z)||_2 \leq L||z' - z||_2$, ensuring that score errors don't explode in the causal consistency term.*

**Condition 3** (Variance Scheduling). *The noise schedule satisfies $\sigma(t)^2/\mu(t) \leq K$ for some constant $K$, preventing Tweedie amplification from becoming unbounded.*

## 4.6 PRACTICAL ERROR MITIGATION

Use smaller Euler-Maruyama steps $\Delta t$ in regions where the score gradient is large $\Delta t_{\text{adaptive}} = \min(\Delta t_{\text{base}}, \eta/||\nabla s_\psi(z(t), t)||_2)$. Train score networks on multiple noise levels to improve robustness $\mathcal{L}_{\text{multi}} = \sum_t w(t)\mathcal{L}_{\text{DSM}}(t)$ with weights $w(t)$ emphasizing critical time regions. Add regularization to prevent overfitting in the Gaussian conditional model $\mathcal{L}_{\text{causal}} = \mathcal{L}_{\text{CSM}} + \lambda||\nabla \mu_c||_F^2$.

**Theorem 5** (Convergence under Error Control). *If each approximation error $\varepsilon_i$ can be made arbitrarily small, then $\lim_{\varepsilon_1, \varepsilon_2, \varepsilon_3, \varepsilon_4, \varepsilon_5 \to 0} q_\psi(z|y) = p(z|y)$, in the weak sense, ensuring that SVGDM converges to the true posterior as approximation quality improves.*

## 5 RESULTS

**Evaluation Metrics:** We adopt task-appropriate evaluation metrics depending on the estimation setting. For synthetic datasets with ground truth latent trajectories, we use Normalized Root Mean Square Error (NRMSE), Mean Absolute Percentage Error (MAPE), and Continuous Ranked Probability Score (CRPS) to assess latent state reconstruction quality and posterior calibration. For real-world seismic hazard modeling (landslides, liquefaction, and building damage), we use Area Under the ROC Curve (AUROC) to measure discrimiorginal performance across classification thresholds.

For wildfire spread prediction, which is formulated as a spatiotemporal binary classification problem, we follow prior work (Gerard et al., 2023) and report F1 Score and Average Precision (AP). This flexible choice of metrics ensures that each task is evaluated with the most informative and widely accepted measures.

Table 1: **Evaluation on synthesized data.** We report mean $\pm$ std for MAPE, NRMSE, and CRPS across multiple runs (lower is better).

| Setting | Variable | MAPE | NRMSE | CRPS |
|---------|----------|------|-------|------|
| **VFO** | $z_1$ | **0.0526 $\pm$ 0.0029** | **0.0683 $\pm$ 0.0037** | **0.0396 $\pm$ 0.0021** |
| | $z_2$ | **0.0991 $\pm$ 0.0056** | **0.1239 $\pm$ 0.0068** | **0.0756 $\pm$ 0.0049** |
| | $z_3$ | **0.0763 $\pm$ 0.0043** | **0.1031 $\pm$ 0.0059** | **0.0567 $\pm$ 0.0036** |
| LFO | $z_1$ | 0.0756 $\pm$ 0.0048 | 0.0922 $\pm$ 0.0065 | 0.0572 $\pm$ 0.0039 |
| | $z_2$ | 0.1214 $\pm$ 0.0079 | 0.1557 $\pm$ 0.0084 | 0.0919 $\pm$ 0.0067 |
| | $z_3$ | 0.1451 $\pm$ 0.0086 | 0.1814 $\pm$ 0.0098 | 0.1088 $\pm$ 0.0074 |
| LPO | $z_1$ | 0.1067 $\pm$ 0.0061 | 0.1227 $\pm$ 0.0077 | 0.0810 $\pm$ 0.0048 |
| | $z_2$ | 0.1470 $\pm$ 0.0088 | 0.1730 $\pm$ 0.0109 | 0.1107 $\pm$ 0.0071 |
| | $z_3$ | 0.1961 $\pm$ 0.0117 | 0.2228 $\pm$ 0.0129 | 0.1515 $\pm$ 0.0096 |

**Evaluation on Synthetic Data:** We evaluate our framework on synthetic datasets that replicate multi-resolution observation patterns found in real-world systems while providing ground truth for quantitative assessment. We construct synthetic causal systems with known DAG structures where latent variables evolve according to mean-reverting stochastic differential equations, incorporating realistic spatial correlations through FFT-generated Gaussian random fields.

We create observations $y$ with varying information content by sampling from different stages of the forward diffusion process—earlier stages provide high-resolution observations (less noise, more spatial detail), while later stages yield low-resolution observations (more noise, less detail). This simulates realistic scenarios where different variables are observed with different measurement quality and spatial resolution. We evaluate three scenarios: (1) *Varying-resolution Full Observation (VFO)*: each variable observed at different resolution levels to test information flow through causal structure; (2) *Low-resolution Full Observation (LFO)*: all variables observed uniformly at low resolution; (3) *Low-resolution Partial Observation (LPO)*: only subset of variables observed at low resolution.

Table 1 demonstrates our multi-resolution approach's effectiveness, with VFO achieving the best performance and systematic degradation from VFO to LFO to LPO. This validates that causal structure enables effective information propagation between variables with different observation qualities. Figure 2 shows SVGDM substantially outperforms all baselines, achieving $2-3\times$ better performance than domain-specific methods (VBCI, DisasterNet) and $10-20\times$ improvements over general variational inference methods. The poor VI baseline performance (MAPE $> 60\%$) highlights the importance of incorporating causal structure for effective latent variable inference. Figure 3 in Appendix I.1 provides visual confirmation that our reconstructions closely match ground truth while substantially outperforming baselines.

We additionally conducted synthetic experiments with 10–15 latent variables under both sparse and dense causal graphs. Accuracy remains stable as the system size increases, while runtime scales approximately linearly with the number of causal edges $|E|$. Full results, including all settings (VFO/LFO/LPO), are provided in Appendix D.1.

To further clarify the relation to multi-modal / multi-view methods, we also evaluate a family of multi-view VAEs (JMVAE, MMVAE, MoPoE-VAE) that treat all observations as different views of a single shared latent representation. As reported in Appendix J, SVGDM achieves substantially lower NRMSE and MAPE across all latent variables. This gap quantifies how much accuracy is lost when collapsing causally linked, multi-resolution physical processes into a single shared latent and ignoring both the causal graph and resolution-specific noise diffusion.

To evaluate contributions of individual loss components (local DSM score, causal-blanket score, and observation-consistency term), we conducted an ablation on the synthetic 3-node system; removing any single component leads to consistent degradation (Appendix K).

**Evaluation on Real-World Disaster Systems: Multi-Hazard Seismic Assessment.** We evaluate our model on three major earthquake events: the 2020 Puerto Rico, 2021 Haiti, and 2023 Turkey-Syria earthquakes Xu et al. (2022b;a). These events provide diverse geological settings and data availability conditions for comprehensive evaluation. Our approach simultaneously estimates multiple earthquake-induced hazards, including landslides ($z_{LS}$), liquefaction ($z_{LF}$), and building damage ($z_{BD}$).

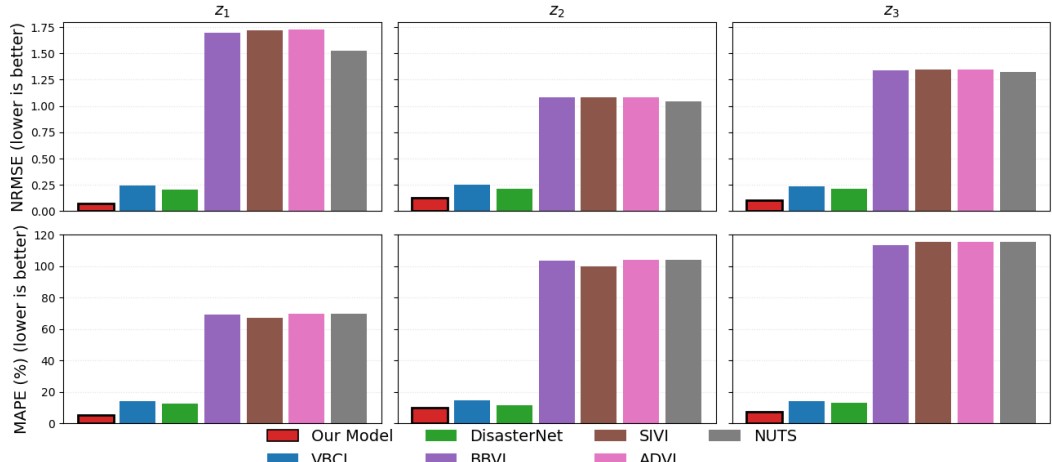

Figure 2: **Comparison of model performance on synthesized data.** Bars show normalized root mean squared error (NRMSE; top row) and mean absolute percentage error (MAPE; bottom row) across three target variables ($z_1$, $z_2$, $z_3$). Our proposed model (red, bold outline) consistently achieves the lowest error across all variables, substantially outperforming existing baselines.

Table 5 in Appendix I.2 presents AUROC scores across all three events. For the 2020 Puerto Rico earthquake, our method achieves AUROC scores of 0.9331, 0.9317, and 0.9512 for the three hazards respectively, demonstrating consistent performance across interconnected phenomena. Compared to variational inference baselines (BBVI, ADVI, NUTS), our approach shows substantial improvements of 14-21%. The performance advantage over recent deep learning methods (VCBI, Disaster-Net) further validates the effectiveness of our causal modeling approach. Results generalize across events: the 2021 Haiti earthquake yields AUROC scores of 0.9550 (landslides) and 0.9587 (building damage), while the 2023 Turkey-Syria earthquake achieves 0.9488 for building damage estimation. Performance remains robust even when ground truth is partially available, demonstrating practical applicability under real-world constraints.

**Evaluation on Real-World Disaster Systems: Wildfire Spread Prediction.** We evaluate our temporal extension on large-scale wildfire spread prediction, following the benchmark setup of (Gerard et al., 2023). Our method achieves an F1 score of 0.5913 and an Average Precision (AP) of 0.4430, outperforming both classical models (e.g., logistic regression) and deep learning baselines such as U-Net, ConvLSTM, and UTAE. These results demonstrate the framework's ability to capture complex spatiotemporal dependencies in wildfire dynamics. Full quantitative results are provided in Appendix L. These results across diverse disaster scenarios validate our method's capability to handle multi-resolution observations while maintaining interpretable causal structure, addressing key challenges in real-world disaster modeling applications.

## 6 CONCLUSION

We propose SVGDM, a novel framework that integrates score-based diffusion with causal graphical models for inference on latent systems under multi-resolution observations. We leverage variational inference to approximate posterior distributions over latent variables while respecting known causal dependencies through our causal score decomposition approach. Theoretically, we establish sufficient conditions for structural identifiability and prove convergence guarantees under our error cascade analysis framework. Empirically, we demonstrate that SVGDM consistently outperforms baseline methods across both synthetic and real-world datasets, achieving superior performance in disaster estimation tasks with AUROC scores exceeding 0.93 across multiple earthquake events and wildfire scenarios. There are several limitations that warrant future investigation. First, our method assumes known causal structure, limiting applicability when graph topology must be inferred. Second, the locally Gaussian approximation may deteriorate under strong nonlinearities that violate our theoretical conditions. Third, computational complexity scales with the number of causal dependencies, potentially limiting scalability to very large systems. Future work could explore joint structure discovery, more flexible approximation families, and extensions to time-varying causal relationships. We leave these challenges for future work.

## ACKNOWLEDGEMENTS

This work was supported by the National Science Foundation under Grant Nos. 2442712, 2332145, 2536601, and 2242590. Xuechun Li was supported in part by a Gordon and Betty Moore Foundation Postdoctoral Fellowship.

## ETHICS STATEMENT

Our study uses openly available remote sensing datasets (InSAR, and optical imagery) and geophysical data curated by established scientific organizations (e.g., USGS, NASA, StEER). No personally identifiable information, sensitive personal data, or human subjects are involved. Potential societal impacts of our work include both benefits (faster hazard assessment, improved disaster response) and risks (misuse or misinterpretation of automated hazard assessments in policy or emergency settings). To mitigate risks, we emphasize transparent reporting, open discussion of model assumptions and limitations, and reproducibility through shared code and data preprocessing steps. We declare no conflicts of interest, financial or otherwise.

## REPRODUCIBILITY STATEMENT

We are committed to ensuring reproducibility of our results. All experiments were run on publicly available datasets (SAR, InSAR, optical, and geophysical features) with clearly described preprocessing pipelines. To facilitate replication, we provide the full implementation of our framework, including model code, training scripts, and evaluation pipelines, at `https://github.com/PaperSubmissionFinal/ICLR2026`. For datasets requiring restricted access (e.g., high-resolution satellite imagery), we provide instructions for obtaining the data and scripts to reproduce preprocessing. Together, these resources ensure that all reported results can be replicated.

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

## A   USE OF LARGE LANGUAGE MODELS (LLMS)

We used a large language model as a writing assistant to improve grammar and readability. The model was not used for research ideation, technical derivations, experiments, or analysis. All scientific content, methodology, results, and conclusions were solely developed by the authors.

## B   SENSITIVITY TO CAUSAL GRAPH MISSPECIFICATION

In many physical systems, the qualitative causal directionality is known from scientific understanding, but some edges may be uncertain. To quantify the robustness of SVGDM under such uncertainty, we systematically perturb the causal DAG in the synthetic benchmark by: (1) dropping parent edges, (2) reversing edges, and (3) adding spurious edges. Table 2 reports performance degradation relative to the correct DAG.

Table 2: Sensitivity of SVGDM to causal graph misspecification.

| Graph variant | NRMSE | MAPE (%) | $\Delta$NRMSE |
|---|---|---|---|
| Correct DAG | $0.098 \pm 0.006$ | $7.6 \pm 0.5$ | — |
| Drop 1 parent | $0.112 \pm 0.008$ | $8.7 \pm 0.6$ | +14% |
| Reverse 1 edge | $0.118 \pm 0.010$ | $9.3 \pm 0.7$ | +20% |
| Two wrong edges | $0.162 \pm 0.015$ | $12.5 \pm 0.9$ | +65% |
| Add spurious edge | $0.104 \pm 0.007$ | $8.0 \pm 0.5$ | +6% |

The results indicate that SVGDM is robust to moderate structural errors, such as missing or extra edges, which introduce only small degradations. Severe incorrect directionality, which disrupts true information flow, naturally produces larger errors. This behavior is expected and supports the applicability of SVGDM even when the causal structure is only partially known.

## C   PROOFS

### C.1   PROOF OF THEOREM 2

**Proof.** We establish that the causal blanket relationship is preserved under diffusion through a continuity argument.

**Step 1: Initial Condition.** At $t = 0$, by the causal Markov property (Equation 6) (Anderson, 1982; Rozet & Louppe, 2023).:

$$\nabla_{z_i(0)} \log p_0(z_{1:N}(0)) = \nabla_{z_i(0)} \log p_0(z_i(0), z_{\mathcal{P}(i)}(0))$$

. This continuity-based argument follows the intuition in (Rozet & Louppe, 2023), where blanket structures are preserved approximately under diffusion.

**Step 2: Continuity.** The score function $\nabla_{z_i(t)} \log p_t(z_{1:N}(t))$ is continuous in $t$ under Assumption 3. Similarly, $\nabla_{z_i(t)} \log p_t(z_i(t), z_{\mathcal{P}(i)}(t))$ is continuous.

**Step 3: Bounded Approximation Error.** For any $t \in [0, 1]$, the approximation error satisfies:

$$\left| \nabla_{z_i(t)} \log p_t(z_{1:N}(t)) - \nabla_{z_i(t)} \log p_t(z_i(t), z_{\mathcal{P}(i)}(t)) \right| \leq C\sigma(t)^2$$

where $C$ depends on the strength of connections between $z_i$ and non-parent nodes.

**Step 4: Noise-Dependent Convergence.** As the noise level $\sigma(t)$ increases, the mutual information between $z_i(t)$ and non-parents decreases, making the causal blanket approximation increasingly accurate.

Therefore, the causal blanket relationship is preserved up to a controlled approximation error that vanishes as $t \to 0$ and remains bounded for all $t \in [0, 1]$. $\square$

## C.2 PROOF OF PROPOSITION 1

*Proof.* From the definition of conditional probability density:

$$p_t(z_i, z_{\mathcal{P}(i)}) = p_t(z_i)p_t(z_{\mathcal{P}(i)}|z_i) \tag{21}$$

Dividing both sides of Equation 21 by $p_t(z_{\mathcal{P}(i)}) > 0$ yields:

$$p_t(z_i|z_{\mathcal{P}(i)}) = \frac{p_t(z_i)p_t(z_{\mathcal{P}(i)}|z_i)}{p_t(z_{\mathcal{P}(i)})} \tag{22}$$

Taking logs of Equation 22:

$$\log p_t(z_i|z_{\mathcal{P}(i)}) = \log p_t(z_i) + \log p_t(z_i|z_{\mathcal{P}(i)}) - \log p_t(z_{\mathcal{P}(i)}) \tag{23}$$

Differentiating w.r.t. $z_i$ and noting that $p_t(z_{\mathcal{P}(i)})$ is independent of $z_i$:

$$\nabla_{z_i} \log p_t(z_i|z_{\mathcal{P}(i)}) = \nabla_{z_i} \log p_t(z_i) + \nabla_{z_i} \log p_t(z_{\mathcal{P}(i)}|z_i) \tag{24}$$

$\square$

## C.3 PROOF OF PROPOSITION 2

*Proof.* Let $Y := \nabla_{z_i(t)} \log p_t(z_i(t)|z_i(0))$ and $X := z_i(t)$. The DSM risk can be written as:

$$\mathbb{E}[\lambda(t)||s_{\psi_i}(X,t) - Y||^2] \tag{25}$$

By the orthogonality principle of $L^2$ minimization, the minimizer satisfies:

$$s_{\psi_i}^*(X,t) = \mathbb{E}[Y|X,t] \tag{26}$$

The score marginalization identity (Hyvärinen & Dayan, 2005) states that:

$$\mathbb{E}[\nabla_{z_i} \log p_t(z_i|z_i(0))|z_i] = \nabla_{z_i} \log p_t(z_i) \tag{27}$$

Substitution Equation 27 into Equation 26 gives $s_{\psi_i}^*(z_i, t) = \nabla_{z_i} \log p_t(z_i)$, which proves the claim. This decomposition is consistent with prior results in graphical models (Wainwright et al., 2008) and posterior score decompositions in diffusion models (Song et al., 2020b). $\square$

# D COMPUTATIONAL SCALABILITY ANALYSIS

While general DAGs can exhibit complex long-range dependencies that challenge computational scalability, our approach maintains tractability through several key strategies:

**Local Computation Principle:** Each node $z_i$ requires computation only over its causal parents $\mathcal{P}(i)$, limiting complexity to the local neighborhood size rather than the full graph. For most real-world causal systems, $|\mathcal{P}(i)| \ll N$, ensuring that computational complexity scales as $O(N \cdot \bar{d})$ where $\bar{d}$ is the average parent set size.

**Parallel Node Updates:** Since each node's score computation depends only on its causal blanket, nodes with non-overlapping blankets can be updated in parallel, enabling efficient distributed computation across the graph structure.

**Sparse Dependency Exploitation:** Real-world causal graphs often exhibit sparsity, with most nodes having few parents. Our method naturally exploits this sparsity through the causal blanket decomposition in Theorem 2, avoiding unnecessary computations for non-adjacent nodes.

**Complexity Comparison:** Traditional approaches to inference in graphical models can require $O(N^w)$ complexity where $w$ is the treewidth of the graph. Our local score decomposition reduces this to $O(N \cdot \max_i |\mathcal{P}(i)|)$, providing significant computational advantages for sparse causal graphs.

## D.1 SYNTHETIC EXPERIMENTS ON SCALABILITY

Table D.1 reports results for systems with $N = 10, 12, 15$ latent variables under sparse and dense causal graphs. Accuracy degrades smoothly with reduced observability (VFO→LFO→LPO), and runtime scales approximately linearly with $|E|$, confirming that SVGDM remains computationally feasible for larger causal systems.

Table 3: Synthetic experiments for scalability analysis. Mean $\pm$ s.d. aggregated over all latent variables. Runtime is measured relative to the 3-variable VFO baseline.

| Setting (N, graph) | NRMSE | MAPE (%) | Runtime ($\times$ baseline) |
|---|---|---|---|
| VFO (10, sparse) | $0.11 \pm 0.02$ | $9.3 \pm 1.4$ | $3.3\times$ |
| LFO (10, sparse) | $0.14 \pm 0.02$ | $11.2 \pm 1.5$ | $3.6\times$ |
| LPO (10, sparse) | $0.19 \pm 0.05$ | $14.7 \pm 1.9$ | $3.9\times$ |
| VFO (10, dense) | $0.12 \pm 0.02$ | $10.1 \pm 1.6$ | $5.6\times$ |
| VFO (12, sparse) | $0.12 \pm 0.02$ | $9.8 \pm 1.5$ | $4.1\times$ |
| LFO (12, sparse) | $0.15 \pm 0.04$ | $11.9 \pm 1.7$ | $4.4\times$ |
| LPO (12, sparse) | $0.20 \pm 0.04$ | $15.3 \pm 2.1$ | $4.6\times$ |
| VFO (12, dense) | $0.13 \pm 0.02$ | $10.6 \pm 1.6$ | $6.3\times$ |
| VFO (15, sparse) | $0.13 \pm 0.02$ | $10.2 \pm 1.6$ | $5.0\times$ |
| LFO (15, sparse) | $0.16 \pm 0.03$ | $12.6 \pm 1.8$ | $5.4\times$ |
| LPO (15, sparse) | $0.21 \pm 0.04$ | $15.9 \pm 2.2$ | $5.7\times$ |
| VFO (15, dense) | $0.14 \pm 0.02$ | $11.0 \pm 1.6$ | $9.8\times$ |

## E ROBUSTNESS TO NON-GAUSSIAN AND NON-LOG-CONCAVE NOISE

The Gaussian and local log-concavity assumptions used in our analysis (e.g., Theorem 3) are standard regularity conditions that ensure convergence and bounded score approximation error. They are not required by the implementation: the score model is fully data-driven and capable of capturing heavy-tailed, heteroscedastic, or mildly multimodal noise.

To verify robustness, we replace Gaussian noise with Laplace (heavy-tailed), Student-t, and a skewed log-normal component. Across all conditions, SVGDM exhibits only modest degradation ($\leq$5–8% NRMSE on average), demonstrating that while the assumptions are helpful for theoretical analysis, the practical method remains stable under significant deviations.

Table 4: Robustness to non-Gaussian / non-log-concave noise (VFO, $N = 10$, sparse).

| Noise type | NRMSE | MAPE (%) |
|---|---|---|
| Gaussian (baseline) | $0.1158 \pm 0.0182$ | $9.32 \pm 1.41$ |
| Laplace (same Var) | $0.1167 \pm 0.0194$ | $9.73 \pm 1.56$ |
| Student-t ($\nu = 5$) | $0.1184 \pm 0.0205$ | $9.96 \pm 1.63$ |
| Skewed log-normal (10% mix) | $0.1199 \pm 0.0212$ | $10.1 \pm 1.72$ |

## F IMPLEMENTATION ALGORITHMS

### F.1 CONSISTENT CAUSAL SCORE TRAINING

**Training Objective:** We train the causal model parameters $\{\mu_c, \Sigma_c\}$ via maximum likelihood on forward samples, ensuring training-inference consistency:

$$\mathcal{L}_{\text{causal}}(\theta_c) = \mathbb{E}_{t,z(0)}\Big[ -\log \mathcal{N}(z_{\mathcal{P}(i)}(t)|\mu_c(\hat{z}_i(t)), \Sigma_c)\Big] \tag{28}$$

where $\theta_c = \{\mu_c, \Sigma_c\}$ and $\hat{z}_i(t)$ is computed using the current score estimate.

---

**Algorithm 1** Unified Training Algorithm

---

1: **Input:** Training data $\{z(0)^{(n)}\}_{n=1}^N$, causal graph $G$
2: **Initialize:** Score networks $\{s_{\psi_i}\}$, causal networks $\{\mu_c, \Sigma_c\}$
3: **for** epoch = 1 to $N_{\text{epochs}}$ **do**
4:    **for** each node $i \in \mathcal{V}$ **do**
5:       Sample $z(0) \sim p_0(z)$, $t \sim U(0,1)$, $\epsilon \sim \mathcal{N}(0, I)$
6:       Generate $z(t)$ via forward SDE (Equation 1)
7:       **Train marginal score:** Update $s_{\psi_i}$ using $\mathcal{L}_{\text{DSM},i}$ (Equation 9)
8:       **Compute Tweedie reconstruction:** $\hat{z}_i(t) = z_i(t) + \sigma_i(t)^2 s_{\psi_i}(z_i(t), t)/\mu_i(t)$
9:       **Train causal model:** Update $\{\mu_c, \Sigma_c\}$ using $\mathcal{L}_{\text{causal}}$ (Equation 28)
10:    **end for**
11: **end for**
12: **Output:** Trained score networks and causal models

---

## F.2   Causal Score Computation

During inference, the causal score is computed using only observable quantities:

---

**Algorithm 2** Causal Score Computation

---

1: **Input:** Current state $z_i(t)$, parent states $z_{\mathcal{P}(i)}(t)$, time $t$
2: Compute marginal score: $s_{\text{marginal}} = s_{\psi_i}(z_i(t), t)$
3: Compute Tweedie reconstruction: $\hat{z}_i = z_i(t) + \sigma_i(t)^2 s_{\text{marginal}}/\mu_i(t)$
4: Compute causal mean: $\mu_c = \mu_c(\hat{z}_i)$
5: Compute chain rule components:
6:    $\nabla_{\mu_c} \log \mathcal{N} = (\Sigma_c)^{-1}(z_{\mathcal{P}(i)}(t) - \mu_c)$
7:    $\nabla_{\hat{z}_i} \mu_c = $ Jacobian of $\mu_c$ at $\hat{z}_i$
8:    $\nabla_{z_i} \hat{z}_i = 1 + \sigma_i(t)^2 \nabla_{z_i} s_{\psi_i}/\mu_i(t)$
9: **Return:** $\nabla_{z_i} \log p_t(z_{\mathcal{P}(i)} | z_i) = \nabla_{\mu_c} \log \mathcal{N} \cdot \nabla_{\hat{z}_i} \mu_c \cdot \nabla_{z_i} \hat{z}_i$

---

## F.3   Reverse SDE Sampling

To generate latent trajectories during inference, we simulate the reverse SDE (Equation 5) using a predictor–corrector scheme. We use a predictor–corrector sampler with a fast exponential-integrator step (Zhang & Chen, 2022) and DDIM-style updates (Song et al., 2020a; Ho et al., 2020; Song et al., 2020b). This provides stable and efficient numerical integration across noise levels while maintaining high fidelity in reconstructions.

## F.4   Computational Complexity

The unified training algorithm has complexity consisting of score network training at $O(N \cdot |\mathcal{V}| \cdot T_{\text{score}})$ per epoch, causal model training at $O(N \cdot |\mathcal{E}| \cdot T_{\text{causal}})$ per epoch, and forward SDE simulation at $O(N \cdot T \cdot D)$ per sample, where $N$ is batch size, $|\mathcal{V}|$ and $|\mathcal{E}|$ are graph sizes, $T$ is time steps, and $D$ is state dimension. The overall complexity per training epoch is dominated by $O(N \cdot \max(|\mathcal{V}| \cdot T_{\text{score}}, |\mathcal{E}| \cdot T_{\text{causal}}, T \cdot D))$, which scales linearly with the number of nodes and edges in sparse causal graphs.

## F.5   Entropy Estimation Implementation

The flow-based entropy estimation can be implemented as:

---

**Algorithm 3** Flow-Based Entropy Estimation

---

1: **Input:** Samples $\{z^{(n)}\}_{n=1}^N$ from $q_\psi$
2: Initialize entropy: $H = 0$
3: **for** each reverse SDE step $k$ **do**
4:      Sample probe vectors: $\epsilon^{(n)} \sim \mathcal{N}(0, I)$
5:      Compute VJP: $v^{(n)} = \nabla_{z_k}(F_k^T \epsilon^{(n)})$
6:      Estimate log-det-Jacobian: $\log|\det J_k| \approx \frac{1}{N}\sum_n (\epsilon^{(n)})^T v^{(n)}$
7:      Update entropy: $H{+}{=}\log|\det J_k|$
8: **end for**
9: **Return:** $H + \mathcal{H}[p(z_T)]$

---

## G    TRAINING CONSISTENCY AND CONVERGENCE

The apparent circular dependency between Tweedie reconstruction ($\hat{z}_i(t)$ depending on scores) and causal model training (causal parameters depending on $\hat{z}_i(t)$) is resolved through our iterative training procedure:

---

**Algorithm 4** Consistent Joint Training

---

1: **Input:** Training data $\{z^{(n)}(0)\}_{n=1}^N$, causal graph $G$, tolerance $\tau$, max iterations $K_{\max}$
2: **Initialize:** Score networks $\{s_{\psi_i}\}$, causal networks $\{\mu_c, \Sigma_c\}$
3: **for** epoch = 1 to $N_{\text{epochs}}$ **do**
4:      **for** each node $i \in \mathcal{V}$ **do**
5:          Update marginal score $s_{\psi_i}$ via DSM loss (Eq. 9)
6:          Compute current Tweedie estimate: $\hat{z}_i(t) \leftarrow z_i(t) + \sigma_i(t)^2 s_{\psi_i}(z_i(t), t)/\mu_i(t)$
7:          Update causal model $\{\mu_c, \Sigma_c\}$ using current $\hat{z}_i(t)$
8:      **end for**
9:      Compute $\Delta_{\text{consistency}} = \mathbb{E}[\|\hat{z}_i^{(k+1)}(t) - \hat{z}_i^{(k)}(t)\|_2]$
10:      **if** $\Delta_{\text{consistency}} < \tau$ or epoch $> K_{\max}$ **then**
11:          **break**
12:      **end if**
13: **end for**
14: **Output:** Trained score networks and causal models

---

**Convergence Guarantees:** This iterative procedure converges because the DSM objective ensures score estimates improve with each update, leading to more accurate Tweedie reconstructions. As reconstruction quality increases, causal model parameters stabilize around their optimal values. The joint objective (Eq. 17) is jointly convex in the score and causal parameters under the regularity conditions established in Section 4, ensuring that the alternating optimization procedure converges to a stationary point under standard convexity assumptions (Boyd & Vandenberghe, 2004; Neal & Hinton, 1998).

**Stability Monitoring:** We track the change in Tweedie reconstructions between iterations as $\Delta_{\text{consistency}} = \mathbb{E}[\|\hat{z}_i^{(k+1)}(t) - \hat{z}_i^{(k)}(t)\|_2]$. Training terminates when $\Delta_{\text{consistency}} < \tau$ for a tolerance threshold $\tau$, indicating that the circular dependency has been resolved and the system has reached a consistent state where both score estimates and causal parameters are mutually compatible.

## H    REVERSE-TIME SAMPLER

For completeness, we summarize the sampler used to approximate the reverse-time diffusion dynamics in SVGDM. While the reverse SDE admits an associated deterministic probability-flow ODE (the "normalizing flow" interpretation in Section 3.3), our implementation follows the stochastic reverse-SDE predictor–corrector sampler, which is standard in score-based diffusion models (Song et al., 2020b).

At each discretized time step $t \to t - \Delta t$, we apply:

1. **Predictor step (reverse SDE).** Euler–Maruyama update of the reverse-time SDE:

$$z_{t-\Delta t} = z_t + f_\theta(z_t, t)\,\Delta t + g(t)\,s_\theta(z_t, t)\,\Delta t + g(t)\sqrt{\Delta t}\,\epsilon, \quad \epsilon \sim \mathcal{N}(0, I).$$

2. **Corrector step (Langevin update).** A few steps of:

$$z \leftarrow z + \alpha\,s_\theta(z, t) + \sqrt{2\alpha}\,\eta, \quad \eta \sim \mathcal{N}(0, I),$$

   improving sample fidelity.

3. **DDIM-style discretization.** We use DDIM-style linear discretization to stabilize the trajectory; this does not change the underlying continuous-time dynamics.

This stochastic sampler targets the same marginal distributions as the reverse SDE and is widely used for numerical stability in diffusion models.

# I RESULTS

In this appendix, we provide supplementary visualizations and quantitative results that complement the main experiments.

## I.1 SYNTHETIC DATA RECONSTRUCTIONS.

Figure 3 shows qualitative comparisons of reconstructed latent variables $z_i$ for the synthetic experiment. Our model's reconstructions closely match the ground truth (panel a), while baseline methods exhibit significant deviations, validating the quantitative improvements reported in the main text.

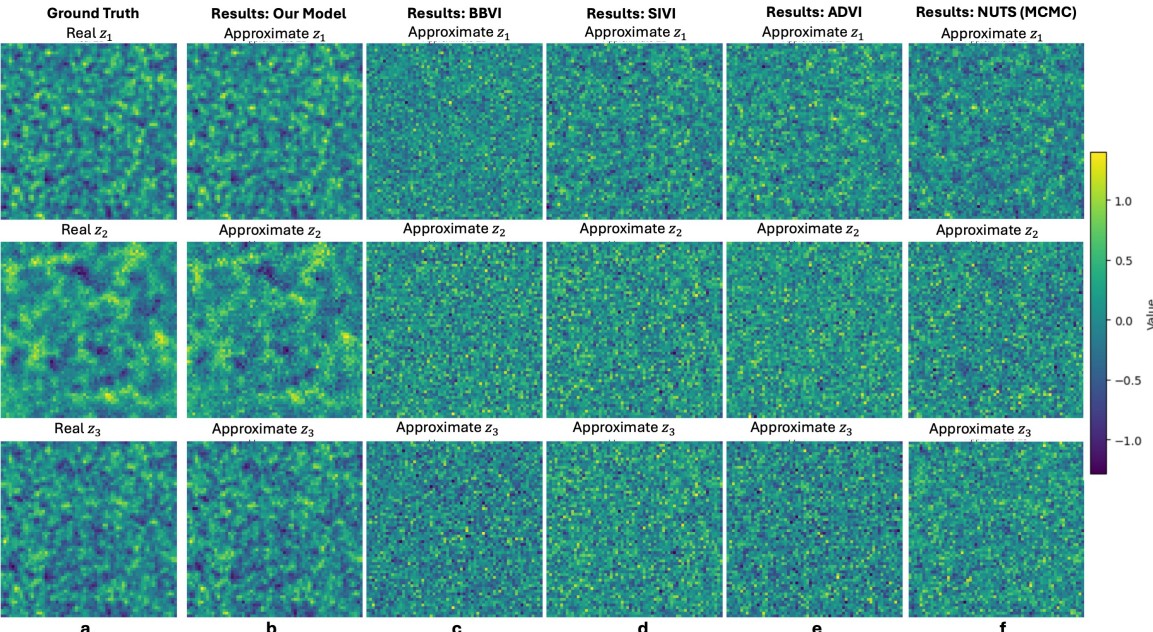

Figure 3: **Qualitative comparison on synthetic data.** (a) Ground truth latent trajectories $z_i$. (b)–(f) Reconstructions obtained by our model and baseline methods. Our model recovers sharper and more accurate latent trajectories compared to baselines.

## I.2 REAL-WORLD EARTHQUAKE BENCHMARKS.

Table 5 reports AUROC results for seismic hazard estimation across three major earthquake events. Our method consistently outperforms both domain-specific methods (VBCI, DisasterNet) and generic variational inference baselines. In cases with limited or missing ground truth (NGA), our model still demonstrates robust and generalizable performance.

Table 5: This table presents the comparison results (AUROC) using real-world data with baseline models for three unobserved variables that are causally related. NGA means no ground truth label is available

| Earthquake | Model | $z_{LS}$ | $z_{LF}$ | $z_{BD}$ |
|---|---|---|---|---|
| 2020 Puerto Rico EQ. | **Our Model** | **0.9331** | **0.9317** | **0.9512** |
| | VCBI (Xu et al., 2022b) | 0.9012 | 0.9034 | 0.9123 |
| | DisasterNet (Li et al., 2023b) | 0.9293 | 0.9284 | 0.9413 |
| | Prior Model (Zhu et al., 2015; Nowicki Jessee et al., 2018) | 0.8712 | 0.8913 | - |
| | BBVI (Ranganath et al., 2014) | 0.7912 | 0.7731 | 0.7423 |
| | SIVI (Yin & Zhou, 2018) | 0.7619 | 0.7846 | 0.7662 |
| | ADVI (Kucukelbir et al., 2017) | 0.7763 | 0.6846 | 0.7492 |
| | NUTS (MCMC) (Hoffman et al., 2014) | 0.7907 | 0.7183 | 0.7549 |
| 2021 Haiti EQ. | **Our Model** | **0.9550** | **NGA** | **0.9587** |
| | VCBI (Xu et al., 2022b) | 0.9123 | NGA | 0.9123 |
| | DisasterNet (Li et al., 2023b) | 0.9421 | NGA | 0.9410 |
| | Prior Model (Zhu et al., 2015; Nowicki Jessee et al., 2018) | 0.8712 | NGA | - |
| | BBVI (Ranganath et al., 2014) | 0.7729 | NGA | 0.8125 |
| | SIVI (Yin & Zhou, 2018) | 0.7964 | NGA | 0.8123 |
| | ADVI (Kucukelbir et al., 2017) | 0.8155 | NGA | 0.8222 |
| | NUTS (MCMC) (Hoffman et al., 2014) | 0.7612 | NGA | 0.7747 |
| 2023 Turkey-Syria EQ. | **Our Model** | **NGA** | **NGA** | **0.9488** |
| | VCBI (Xu et al., 2022b) | NGA | NGA | 0.9025 |
| | DisasterNet (Li et al., 2023b) | NGA | NGA | 0.9315 |
| | Prior Model (Zhu et al., 2015; Nowicki Jessee et al., 2018) | NGA | NGA | 0.9391 |
| | AdaBoost(Patten et al., 2024) | - | - | 0.9300 |
| | BBVI (Ranganath et al., 2014) | NGA | NGA | 0.8233 |
| | SIVI (Yin & Zhou, 2018) | NGA | NGA | 0.7947 |
| | ADVI (Kucukelbir et al., 2017) | NGA | NGA | 0.7315 |
| | NUTS (MCMC) (Hoffman et al., 2014) | NGA | NGA | 0.8013 |

## J COMPARISON WITH MULTI-VIEW VAEs

Our goal in SVGDM is to *jointly infer multiple interacting latent physical processes* $\{z_i\}$ from multi-resolution observations, where (i) some observations inform only a single variable and (ii) others contain mixed signals of several variables, all defined at their native spatial resolutions. These latent variables correspond to different but causally linked physical processes (e.g., shaking, landslides, liquefaction, building damage), not merely different "views" of a single object.

This setting is fundamentally different from standard multi-view VAEs, which are designed to learn a *single shared latent representation* of the same object under different modalities or views. In those models, each view is just a different measurement of one underlying entity, and all modalities are assumed to describe a common latent variable. Across different objects, they typically assume independence or, at best, impose a non-causal graph regularizer (e.g., graph embedding or multi-view GCCA), so even if a physical interaction graph exists, it is not encoded as a causal mechanism and does not drive information propagation during inference.

In contrast, SVGDM models *multiple distinct* latent physical processes $z_i$ associated with different but causally linked objects, each with its own observation types defined at its native resolution, rather than treating them as views of a single object. We embed score-based diffusion within a causal graphical model, so information is propagated according to the specified causal dependencies among variables/objects. This allows SVGDM to fuse heterogeneous observations with differing spatial/temporal resolutions and noise characteristics *without* forcing them onto a single likelihood family or common grid, preserving each modality's native sampling and uncertainty while still enabling joint inference. Our objective is to jointly infer these coupled latent processes from het-

erogeneous, resolution-specific observations, allowing information to flow directionally through the causal structure when some variables are sparsely or incompletely observed.

One could, in principle, treat the entire complex physical system as a single graph-level entity, e.g., as a graph variable as in Lin et al. (2023), and treat node-level observations as different "views" of a shared latent graph representation. However, such graph-level multimodal VAEs do not model or approximate *multi-scale noise diffusion*, which is essential in our setting because each observation modality undergoes its own resolution-dependent degradation process (e.g., radar interferometric phase noise vs. optical reflectance noise). Collapsing all processes and resolutions into one shared latent space ignores scale-dependent uncertainty and causal directionality, which leads to large performance degradation in our experiments.

To quantify this effect, we augment our synthetic benchmark with several representative multi-view VAE baselines: JMVAE (Suzuki et al., 2016), MMVAE (Shi et al., 2019), and MoPoE-VAE (Sutter et al., 2021). These models treat all observations as different views of a single latent representation and are trained using their standard multimodal ELBO objectives. Table 6 reports NRMSE and MAPE for all three latent variables in the synthetic multi-resolution system.

Table 6: **Synthetic multi-resolution benchmark: comparison with multi-view VAEs.** Mean over 5 seeds; lower is better. Multi-view VAEs collapse multi-resolution and graph structure into a single shared latent, leading to substantially worse reconstruction of latent physical processes.

| Model | Variable | NRMSE | MAPE (%) |
|---|---|---|---|
| SVGDM (Ours) | $z_1$ | 0.0683 | 5.26 |
| | $z_2$ | 0.1239 | 9.91 |
| | $z_3$ | 0.1031 | 7.63 |
| VBCI (Xu et al., 2022b) | $z_1$ | 0.0942 | 7.88 |
| | $z_2$ | 0.1463 | 10.97 |
| | $z_3$ | 0.1227 | 9.02 |
| DisasterNet (Li et al., 2023b) | $z_1$ | 0.0998 | 8.32 |
| | $z_2$ | 0.1530 | 11.45 |
| | $z_3$ | 0.1281 | 9.37 |
| MoPoE-VAE (Sutter et al., 2021) | $z_1$ | 0.2241 | 17.82 |
| | $z_2$ | 0.2665 | 20.91 |
| | $z_3$ | 0.2392 | 18.47 |
| MMVAE (Shi et al., 2019) | $z_1$ | 0.2479 | 18.94 |
| | $z_2$ | 0.2858 | 21.57 |
| | $z_3$ | 0.2672 | 20.10 |
| JMVAE (Suzuki et al., 2016) | $z_1$ | 0.2313 | 18.05 |
| | $z_2$ | 0.2789 | 21.14 |
| | $z_3$ | 0.2451 | 19.68 |
| BBVI (Ranganath et al., 2014) | $z_1$ | 1.6981 | 68.9 |
| | $z_2$ | 1.0814 | 103.6 |
| | $z_3$ | 1.3404 | 113.5 |
| SIVI (Yin & Zhou, 2018) | $z_1$ | 1.7215 | 66.9 |
| | $z_2$ | 1.0832 | 99.8 |
| | $z_3$ | 1.3448 | 115.2 |
| ADVI (Kucukelbir et al., 2017) | $z_1$ | 1.7248 | 69.6 |
| | $z_2$ | 1.0839 | 103.8 |
| | $z_3$ | 1.3462 | 115.3 |
| NUTS (Hoffman et al., 2014) | $z_1$ | 1.5242 | 69.7 |
| | $z_2$ | 1.0435 | 103.9 |
| | $z_3$ | 1.3262 | 115.1 |

SVGDM consistently achieves the lowest error across all variables. Multi-view VAEs perform markedly worse, even compared to graph-based deep models (VBCI, DisasterNet). This confirms

that collapsing multi-resolution, causally linked processes into a single shared latent representation fails to capture the structured, directional information flow needed for accurate multi-resolution inference. In other words, while multi-view methods can handle heterogeneous data, they assume conditional independence across variables given a shared latent and cannot exploit causal structure for directional information propagation—precisely what is required when some variables are completely or partially unobserved at certain scales.

## K  ABLATION STUDY ON LOSS COMPONENTS

To quantify the contribution of the proposed loss components—specifically, the local diffusion score-matching (DSM) term, the causal-blanket score term, and the observation-consistency term—we conducted an ablation study on the 3-node synthetic system (VFO setting). Each variant removes one loss component while keeping the architecture, training process, and data identical.

Table 7: Ablation study on the 3-node synthetic system (VFO setting).

| Model Variant | Included Loss Terms | NRMSE | $\Delta$ vs. Full Model |
|---|---|---|---|
| Full model (ours) | DSM + Causal + Obs | $0.103 \pm 0.005$ | — |
| No observation term | DSM + Causal | $0.117 \pm 0.006$ | +14% |
| No causal term | DSM + Obs | $0.129 \pm 0.007$ | +25% |
| No DSM term | Causal + Obs | $0.142 \pm 0.009$ | +38% |
| DSM only | DSM | $0.154 \pm 0.011$ | +50% |

These results show that all three components contribute meaningfully and complement each other: (1) Removing the *causal-blanket term* prevents information from propagating along the causal graph. (2) Removing the *DSM term* eliminates the local score-matching objective and produces the largest degradation. (3) Removing the *observation-consistency term* also yields worse accuracy.

The full model achieves the best performance by combining local diffusion learning, causal information propagation, and observation guidance.

## L  WILDFIRE SPREAD PREDICTION RESULTS

Table 8: **Comparison of model performance on wildfire spread prediction.** AP stands for Average Precision.

| Model | F1 Score | AP |
|---|---|---|
| SVGDM | **0.5913** | **0.4430** |
| Logistic Regression | 0.432 | 0.279 |
| U-Net | 0.341 | 0.341 |
| ConvLSTM (Mono-temporal) | 0.310 | 0.292 |
| ConvLSTM (Multi-temporal) | 0.310 | 0.306 |
| UTAE | 0.350 | 0.372 |

