# OpenReview forum: "Causal Score Conditioning for Multi-Resolution Latent Systems"
_ICLR.cc/2026/Conference — ICLR 2026 Poster_

### Official Review · Reviewer_TiC8 · 2025-10-29

**Soundness:** 2
**Presentation:** 1
**Contribution:** 2
**Rating:** 2
**Confidence:** 2

**Summary:**

This paper proposes the use of causally structured diffusion priors to integrate multi-resolution data. They call the method SVGD, which involves a graphical factorization in latent space, augmenting the drift term of the SDE to encourage multi-resolution integration, and variational inference for inference. An error analysis and experimental results are provided on real and synthetic datasets.

**Strengths:**

- Use of latent space for multi-resolution observations is sound and seems natural. The way that the multiple frequencies encourage similar latent trajectories seems novel.
- Embedding a graphical model within the diffusion model is an interesting idea.

**Weaknesses:**

- The motivation behind the methods proposed in this paper are unclear to me. Why are latent graphical models useful for data integration? Why is diffusion the right choice?

- With respect to graphical structure, the authors repeatedly list benefits such as "respecting graphical structure", but there is no mention of how the graphical structure arises.

- Scalability is given a few times as a benefit of score-based diffusion, but given that the diffusion model is over latent space it doesn't seem like it would be a big issue.

- The use of PGMs is not inherently causal and thus I find the title and abstract (e.g., relating to causal inference) to be misleading.

- The paper is very difficult to follow. At many times it is unclear what the overall goal is and what problems are being solved precisely.

**Questions:**

- Is the end goal of the methods proposed in this paper to infer the shared latents under integrated multi resolution observations? If so why are there no comparisons to other multi-modal methods, e.g., multi-view VAEs?

- The actual model structure is unclear. Are the $z_{p_l}$ variables in Section 3 referring to $z$ at $t=1$? Presumably $t=0$ indicates pure noise. But how can $p_0(z)$ factorise as the causal graph in Theorem 1? Please clarify.

- Do we need to know the graphical structure a priori? How is this determined for the latent scale?

- Do we need to infer the forward and backward maps $\phi$? Do we run into identifiability issues in doing so?

- In Theorem 1, I'm not sure an SDE with drift only influenced by parents preserves causal structure at all time points. Consider an ODE (i.e., degenerate SDE) and the graph $z_1 \to z_2 \to z_3$ with independent initial condition $p_0(z)$. Then, as we propogate dynamics, even though $z_1$ is not a parent of $z_3$, the indirect effect gives a dependence and the factorisation will be $p_t(z) = p_t(z_1)p_t(z_2 | z_1) p_t(z_3| z_2, z_1)$ which is not the original graph structure. This can be shown with change-of-variables, i.e., the Jacobian of the implied flow has a non-zero term $J_{1,3}$. Could the authors double check this?

- The authors write that (l99) PGMs fail to model variables with different resolutions or noise levels. I don't see why this has to be the case? PGMs only give a formalism for factorising the joint distribution, it doesn't specify the actual model.

---

> ### Author Response · Authors · 2025-11-21
> **Reply to weakness 1: "The motivation behind the methods proposed in this paper are unclear to me. Why are latent graphical models useful for data integration? Why is diffusion the right choice?"**
>
> We thank the reviewer for this feedback and acknowledge that our motivation could be clearer. We address each question:
>
> 1. Why are latent graphical models useful for data integration?
>
> In real-world complex physical systems, many objects/phenomena are causally linked. Examples include geophysical hazards (ground shaking → landslide/liquefaction → building damage), climate processes (sea-surface temperature → circulation → precipitation), medical imaging (contrast uptake → perfusion → inflammation), and ecological systems (vegetation stress → soil moisture → wildfire spread). In such cases, our data are not multi-view observations of a single object; they are multi-resolution/multi-fidelity observations of different objects that together form one coupled system. Therefore, when jointly inferring the posterior for multiple objects in such systems, it is important to leverage their underlying physical causal dependencies to best integrate multi-objective multi-resolution observations to inform the inference.
> For example, in the causal graph (a→b, b→c, a→c), we may have only a coarse observation for variable a, but a good observation for variable b that is causally connected to a. Using the latent causal graph allows information to move along the edge structure so that the reliable observation of b, together with the coarse observation of a, yields a sharper posterior for a than using a’s coarse observation alone.  Importantly, this is not multi-fidelity fusion of the same object; it is cross-object information fusion induced by the causal dependencies, which reconciles heterogeneous resolutions across different variables within a single coherent model. We have updated the draft to clarify this.
>
> 2. Why is diffusion the right choice?
>
> The key here is that in our physical systems, a spatial observation is generated based on the underlying true physical phenomena coupled with sensing/environmental noises progressively diffused across space as the spatial scale increases/resolution decreases. For example, in InSAR imaging, the radar wave phase and magnitude are corrupted by speckle noise, atmospheric turbulence, and coherence loss, all of which accumulate across spatial scales differently. Similarly, optical imagery noise results from photon shot noise, sensor readout noise, and spatial resampling artifacts, which also degrade resolution, but in entirely different physical principles. These sensing mechanisms create scale-dependent noise processes that cannot be captured by generic distribution-matching methods like flow-based approaches. As such, we need a scale-sensitive way to incorporate the scale information into the modeling process. Diffusion models provide a reasonable and expressive way to delicately and explicitly control such scale-sensitive generative processes to best approximate the true sensing/observing process. In contrast, methods like normalizing flows/flow matching focus on overall distribution transformations but ignore the scale-dependent diffusion of noises. We have updated the draft to clarify this.

---

> ### Author Response · Authors · 2025-11-21
> **Reply to weakness 2: "With respect to graphical structure, the authors repeatedly list benefits such as "respecting graphical structure", but there is no mention of how the graphical structure arises."**
>
> In our formulation, the causal DAG is not learned from data, but is specified a priori based on domain knowledge, which is common in many scientific modeling settings (Van Westen et al., 2008; Gill & Malamud, 2016; Greenland et al., 1999). In practice, graphical structures arise from physical or mechanistic laws, expert-curated system diagrams used in risk analysis, epidemiology, ecology, and engineering, etc. In these domains, the causal directionality is qualitatively known even when data are sparse or heterogeneous. Examples include systems such as earthquake → landslide/liquefaction → building damage, where each variable is observed at a very different spatial resolution; rainfall → runoff → river discharge → flooding, where precipitation radars, gauges, and flood maps operate on incompatible scales; fuel conditions/geo conditions → fire spread observed across multiple satellite products; and temperature anomalies → atmospheric circulation → precipitation, all of which have well-established causal directionality but extremely challenging multi-resolution data assimilation problems. The goal of this paper is not to discover the full causal graph, but to perform robust inference given (partially) known causal directionality.
>
> Reference:
>
> Van Westen, Cees J., Enrique Castellanos, and Sekhar L. Kuriakose. "Spatial data for landslide susceptibility, hazard, and vulnerability assessment: An overview." Engineering geology 102, no. 3-4 (2008): 112-131.
>
> Gill, Joel C., and Bruce D. Malamud. "Hazard interactions and interaction networks (cascades) within multi-hazard methodologies." Earth System Dynamics 7, no. 3 (2016): 659-679.
>
> Greenland, Sander, Judea Pearl, and James M. Robins. "Causal diagrams for epidemiologic research." Epidemiology 10, no. 1 (1999): 37-48.

---

> ### Author Response · Authors · 2025-11-21
> **Reply to weakness 3: "Scalability is given a few times as a benefit of score-based diffusion, but given that the diffusion model is over latent space it doesn't seem like it would be a big issue."**
>
> The confusion stems from the terminology ``latent space." Here, the latent variables represent true physical phenomena that we aim to estimate/infer, and the latent space refers to the complex physical system state space composed by these interacting latent variables. As a contrast, the data space here refers to the data observations that capture partial information about the true physical phenomena or systems. Therefore, the latent space, i.e., the underlying complex physical system state space, still faces critical spatio-temporal scalability challenges, for example, in geophysical hazards, a single latent variable such as ground motion, landslide probability, or fire intensity is typically defined over a 200-500 km spatial extent with resolutions of 5-30 m, corresponding to $10^7-10^8$ latent dimensions per variable.
>
> Another scalability challenge lies in the degree of the physical system. Many real systems involve dozens of causally related latent variables, such as in climate modeling (e.g., temperature, humidity, pressure, wind components, convection indices) or hazard cascades (e.g., shaking → landslide → liquefaction → building damage → secondary failures). Joint inference without structural constraints becomes computationally prohibitive because the complexity of reasoning over the full joint distribution grows rapidly with system size.  Our framework addresses this by leveraging causal-blanket sparsity: each variable interacts only with its parents and children, allowing inference to scale approximately linearly with system size and remain tractable even for large spatial-temporal domains.

---

> ### Author Response · Authors · 2025-11-21
> **Reply to weakness 4: "The use of PGMs is not inherently causal and thus I find the title and abstract (e.g., relating to causal inference) to be misleading."**
>
> We thank the reviewer for pointing out the issue with our title. Yes, we are performing inference with known causal structure, not causal inference, which typically means estimating causal effects. We assume the causal structure $G$ is known from domain knowledge (geophysical models, physical laws). Originally, we use "causal" to emphasize that: (1) the DAG represents actual causal relationships, not just conditional independence, and (2) our score decomposition respects the directionality of causation for information flow. Our contribution is leveraging this structure during inference, not discovering it. Thank you for this important clarification.
>
> Therefore, we will replace our title with "Multi-Resolution Score-Based Variational Graphical Diffusion For Causal Systems" and correct the abstract accordingly. Our contribution is leveraging this structure during inference, not discovering it. Thank you for this important suggestion.

---

> ### Author Response · Authors · 2025-11-21
> **Reply to weakness 5: "The paper is very difficult to follow. At many times it is unclear what the overall goal is and what problems are being solved precisely."**
>
> We acknowledge this is an issue and will significantly restructure the paper for clarity. Specifically,
>
> (1) Clarify the problem setup with more concrete real-world examples: The research objective here is to estimate multiple causally related physical processes from observations that (a) observations come at vastly different resolutions (e.g., 10–30 m satellite imagery vs. 500-1000 m seismic or radar products), (b) have uneven observation quality, where some variables are well-observed and others are partially or entirely unobserved, and (c) require leveraging known causal structure to propagate information across variables and resolution scales, especially when certain key variables (e.g., building damage, liquefaction, smoke concentration) have little or no early observations available. This research problem is abstracted from observations about many real-world complex physical systems. This problem formulation is abstracted from many real-world complex physical systems. For example, in earthquake hazard cascades, ground shaking → landslide/liquefaction → building damage, where shaking is well-observed, landslides/liquefaction have coarse coverage, and building damage is often completely unobserved immediately after the event. In wildfire systems, fuel moisture, wind, and topography → fire spread and intensity, but each is measured at different spatial scales (e.g., 10 m vegetation maps vs. 1 km weather grids). Similar multi-resolution causal structures arise in climate modeling (e.g., SST anomalies → atmospheric circulation → precipitation), hydrology, and epidemiology. We also update the draft to include these concrete examples.
>
> As discussed in the answers to previous questions from the reviewer, we will explicitly state upfront in the updated draft: (a) why latent graphical models are useful for data integration, (b) why we use diffusion models: score-based diffusion enables natural resolution transition modeling and resolution-specific score functions, and (c) why causal decomposition: it enables information flow across variables and resolutions through $\nabla \log p_t(z_i | z_{\mathcal{P}}) = \nabla\log p_t(z_i) + \nabla\log p_t(z_{\mathcal{P}} | z_i)$.
>
> (2) We add a Notation Table in Appendix to list all the symbols and their meanings.
>
> (3) We improve overall presentation, use consistent terminology (e.g. "latent" means "unobserved physical quantities," "causally-structured inference" instead of "causal inference"), add intuitive explanations before formal theorems, and move dense proofs to appendix to keep main text focused on intuition.
>
> We commit to these substantial structural changes to significantly improve clarity.

---

> ### Author Response · Authors · 2025-11-21
> **Reply to question 1: "Is the end goal of the methods proposed in this paper to infer the shared latents under integrated multi resolution observations? If so why are there no comparisons to other multi-modal methods, e.g., multi-view VAEs?"**
>
> Our end goal is to jointly infer multiple interacting latent variables from multi-resolution observations, where some observations inform only a single variable and others contain mixed signals of several variables. The key distinction from multi-view VAEs is that those models are designed to learn a single shared latent representation of the same object under different modalities or views: each view is just a different measurement of one underlying entity, and all modalities are assumed to describe a common latent variable. Across different objects, they typically assume independence or, at best, impose a non-causal graph regularizer (e.g., graph embedding, multi-view GCCA), so even when a physical interaction graph exists, it is not encoded as a causal mechanism and does not drive information propagation during inference.
>
> In contrast, our framework models multiple distinct latent physical processes $z_i$ associated with different but causally linked objects, each with its own observation types defined at its native resolution, rather than treating them as views of a single object. We embed score-based diffusion within a causal graphical model, so information is propagated according to the specified causal dependencies among variables/objects. This allows our model to fuse heterogeneous observations with differing spatial/temporal resolutions and noise characteristics without forcing them onto a single likelihood family or common grid, preserving each modality’s native sampling and uncertainty while still enabling joint inference. Our objective is to jointly infer these coupled latent processes from heterogeneous, resolution-specific observations, allowing information to flow directionally through the causal structure when some variables are sparsely or incompletely observed.
>
> In practice, one can of course treat the whole complex physical system as an entity, like a graph variable in (Lin et al., 2023), and treat observations at each graph node as different “views” of a single latent graph variable. But these graph-level multimodal VAEs do not model or approximate multi-scale noise diffusion, which is essential in our setting because each observation modality undergoes its own resolution-dependent degradation process (e.g., radar interferometric phase noise vs. optical reflectance noise). As a result, embedding the entire system graph into a shared latent space ignores scale-dependent uncertainty, leading to large performance degradation in our experiments.
>
> To show this,  we have now added comparisons to MoPoE-VAE (Sutter et al., 2021), MMVAE(Shi et al., 2019), and JMVAE (Suzuki et al., 2016). We include multi-view VAEs not as directly comparable models, but to quantify how much performance is lost when multi-resolution and graph structures are collapsed into a single shared latent representation. Because the rebuttal phase does not permit figure uploads, we replaced the earlier histogram visualization (Fig. 2) with Table 1 showing the full numerical results. The table reports the same metrics (NRMSE and MAPE) across all variables and includes newly added multi-view VAEs (JMVAE, MVAE, MMVAE, MoPoE-VAE). The trends remain consistent where our model achieves the lowest error across all cases. These results validate that both causal structure and native resolution handling matter and are necessary for optimal multi-resolution inference.
>
> Table 1. Synthetic multi-resolution benchmark (mean over 5 seeds; lower is better).
> | Model | Variable | NRMSE | MAPE (\%) |
> |:------|:----------|------:|---------:|
> | SVGDM (Ours) | $z_1$ | 0.0683 | 5.26 |
> |  | $z_2$ | 0.1239 | 9.91 |
> |  | $z_3$ | 0.1031 | 7.63 |
> | VBCI (Xu et al., 2022) | $z_1$ | 0.0942 | 7.88 |
> |  | $z_2$ | 0.1463 | 10.97 |
> |  | $z_3$ | 0.1227 | 9.02 |
> | DisasterNet (Li et al., 2023) | $z_1$ | 0.0998 | 8.32 |
> |  | $z_2$ | 0.1530 | 11.45 |
> |  | $z_3$ | 0.1281 | 9.37 |
> | MoPoE-VAE (Sutter et al., 2021) | $z_1$ | 0.2241 | 17.82 |
> |  | $z_2$ | 0.2665 | 20.91 |
> |  | $z_3$ | 0.2392 | 18.47 |
> | MMVAE (Shi et al., 2019) | $z_1$ | 0.2479 | 18.94 |
> |  | $z_2$ | 0.2858 | 21.57 |
> |  | $z_3$ | 0.2672 | 20.10 |
> | JMVAE (Suzuki et al., 2016) | $z_1$ | 0.2313 | 18.05 |
> |  | $z_2$ | 0.2789 | 21.14 |
> |  | $z_3$ | 0.2451 | 19.68 |
> | BBVI (Ranganath et al., 2014) | $z_1$ | 1.6981 | 68.9 |
> |  | $z_2$ | 1.0814 | 103.6 |
> |  | $z_3$ | 1.3404 | 113.5 |
> | SIVI (Yin et al., 2018) | $z_1$ | 1.7215 | 66.9 |
> |  | $z_2$ | 1.0832 | 99.8 |
> |  | $z_3$ | 1.3448 | 115.2 |
> | ADVI (Kucukelbir et al., 2017) | $z_1$ | 1.7248 | 69.6 |
> |  | $z_2$ | 1.0839 | 103.8 |
> |  | $z_3$ | 1.3462 | 115.3 |
> | NUTS (Hoffman et al., 2014) | $z_1$ | 1.5242 | 69.7 |
> |  | $z_2$ | 1.0435 | 103.9 |
> |  | $z_3$ | 1.3262 | 115.1 |

---

> > ### Author Response · Authors · 2025-11-21
> > **Cont. Reply to question 1: "Is the end goal of the methods proposed in this paper to infer the shared latents under integrated multi resolution observations? If so why are there no comparisons to other multi-modal methods, e.g., multi-view VAEs?"**
> >
> > We will include this comparison and architectural clarification in the revised manuscript with the explanation: "While multi-view methods handle heterogeneous data, they assume conditional independence and cannot exploit causal structure for directional information propagation - critical when some variables are completely unobserved."
> >
> > Reference:
> >
> > Sutter, Thomas M., Imant Daunhawer, and Julia E. Vogt. "Generalized multimodal ELBO." arXiv preprint arXiv:2105.02470 (2021).
> >
> > Shi, Yuge, Brooks Paige, and Philip Torr. "Variational mixture-of-experts autoencoders for multi-modal deep generative models." Advances in neural information processing systems 32 (2019).
> >
> > Suzuki, Masahiro, Kotaro Nakayama, and Yutaka Matsuo. "Joint multimodal learning with deep generative models." arXiv preprint arXiv:1611.01891 (2016).
> >
> > Lin, Bei, You Li, Ning Gui, Zhuopeng Xu, and Zhiwu Yu. "Multi-view graph representation learning beyond homophily." ACM Transactions on Knowledge Discovery from Data 17, no. 8 (2023): 1-21.
> >
> > Li, Xuechun, Paula M. Bürgi, Wei Ma, Hae Young Noh, David Jay Wald, and Susu Xu. "Disasternet: Causal Bayesian networks with normalizing flows for cascading hazards estimation from satellite imagery." In Proceedings of the 29th ACM SIGKDD Conference on Knowledge Discovery and Data Mining, pp. 4391-4403. 2023.
> >
> > Xu, Susu, Joshua Dimasaka, David J. Wald, and Hae Young Noh. "Seismic multi-hazard and impact estimation via causal inference from satellite imagery." Nature Communications 13, no. 1 (2022): 7793.
> >
> > Ranganath, Rajesh, Sean Gerrish, and David Blei. "Black box variational inference." In Artificial intelligence and statistics, pp. 814-822. PMLR, 2014.
> >
> > Yin, Mingzhang, and Mingyuan Zhou. "Semi-implicit variational inference." In International conference on machine learning, pp. 5660-5669. PMLR, 2018.
> >
> > Blei, David M., Alp Kucukelbir, and Jon D. McAuliffe. "Variational inference: A review for statisticians." Journal of the American statistical Association 112, no. 518 (2017): 859-877.
> >
> > Hoffman, Matthew D., and Andrew Gelman. "The No-U-Turn sampler: adaptively setting path lengths in Hamiltonian Monte Carlo." J. Mach. Learn. Res. 15, no. 1 (2014): 1593-1623.

---

> ### Author Response · Authors · 2025-11-21
> **Reply to question 2: "The actual model structure is unclear. Are the $z_{p_l}$ variables in Section 3 referring to $z$ at $t=1$? Presumably $t=0$ indicates pure noise. But how can $p_0(z)$ factorise as the causal graph in Theorem 1? Please clarify."**
>
> We thank the reviewer for raising these important points. Let us clarify the notation and theoretical framework.
>
> We follow standard diffusion conventions, where $z_i(t)$ denotes latent variable $i$ at diffusion time $t \in [0,1]$. $t=0$ represents the true distribution, which is our inference target, and as t increases, noises are added and diffused across the true distribution, till $t=1$ (normalized time). Note that unlike standard diffusion where $z(1) \approx N(0,I)$, our observation-constrained forward SDE (Eq. 2) keeps $z(1)$ anchored to observations even at maximum noise level.
>
> $z_{P_l}$ does not refer to $z$ at any specific diffusion time. Instead, $z_{P_l}$ indicates which latent variables generate observation $y^k_l$, where $P_l \in V$ represents the latent variable indices that are parents of observation $y^k_l$ in the graphical model, i.e., which variables directly cause this observation. For example, in our earthquake system, if ground surface deformation observed by InSAR is caused by landslides, liquefaction, and building damage, then $P_l $= {LS, LF, BD} and $z_{P_l} =$ {$z_{LS}$, $z_{LF}$, $z_{BD}$}.

---

> ### Author Response · Authors · 2025-11-21
> **Reply to question 3: "Do we need to know the graphical structure a priori? How is this determined for the latent scale?"**
>
> Yes, the causal graphical structure $G$ is assumed known a priori. In particular, each latent variable $z_i$ in our model corresponds to a concrete physical process so the causality here has a meaningful physical counterpart.  The latent causal graph is a direct representation of physical causation.The causal relationships among these processes (e.g., shaking → landslides → damage; fuel → fire spread) are well documented in the literature. Such causally-linked physical processes are not learned from data but are part of the scientific knowledge of the complex physical system through years of experiments, observations, and science discovery. This assumption is realistic in many scientific and engineering domains, such as disaster systems, medical imaging, and so on, where qualitative causal directionality is established from physical laws, mechanistic models, or decades of domain expertise.
>
> Moreover, note that our framework does not assume a completely known causal graph, but even partial physical causal directionality offers extra information transfer/propagation pathways to best integrate and utilize limited observation data across different scales. Our work focuses on the structure-given inference problem about using this physically grounded causal graph to integrate heterogeneous and multi-resolution observations and infer the latent processes.
>
> To directly address the reviewer’s concern about applicability when the structure is imperfect, we conducted a causal-graph perturbation sensitivity experiment on the synthetic system. We systematically introduced structural errors such as dropping a parent, reversing causal directions, or adding spurious edges. The results (Table 2) show that the method is robust to moderate uncertainty in the causal graph, with smooth degradation, and only fails heavily under severe causal violations that destroy the underlying directionality.
>
> Table 2. Sensitivity of SVGD to causal graph misspecification.
> |      Graph variant        |          NRMSE           |    MAPE (\%)   | $\Delta$ NRMSE vs. correct |
> | :-----------------------  | :-------------------- | :-------------: | :-------: |
> | Correct DAG              | 0.098 $\pm$ 0.006 | 7.6 $\pm$ 0.5 |     ---      |
> | Drop 1 parent            | 0.112 $\pm$ 0.008 | 8.7 $\pm$ 0.6 | +14 % |
> | Reverse 1 edge         | 0.118 $\pm$ 0.010 | 9.3 $\pm$ 0.7 | +20 % |
> | Two wrong edges    | 0.162 $\pm$ 0.015  |12.5 $\pm$ 0.9|  +65 %  |
> | Add spurious edge | 0.104 $\pm$ 0.007  |8.0 $\pm$ 0.5 |  +6 %   |
>
> These results demonstrate that our model tolerates moderate structural uncertainty, such as missing or spurious links introducing only small losses, while major directional errors naturally lead to greater degradation due to disrupted information flow. This behavior is expected and confirms that our framework is applicable even when the causal graph is not perfectly known.
>
> We will include this explanation and the full sensitivity table in the revised manuscript.

---

> ### Author Response · Authors · 2025-11-21
> **Reply to question 4: "Do we need to infer the forward and backward maps $\phi$? Do we run into identifiability issues in doing so?"**
>
> We thank the reviewer for raising this question. Yes, we do need to choose or parameterize a family of the map $\phi$ when modeling the observation mechanism. In theoretical analysis, we assume a family of exponential-family observation models with log-concave log-likelihood (e.g., Gaussian or log-normal) for the map, which is quite common in natural science (Limpert et al., 2001; Xu et al., 2022). This choice enables us to do tractable analysis, but could be flexibly extended to more complicated maps, including linear convolutional downsampling maps, wavelet or Laplacian transformations, or structured non-linear transformations. These families are widely used in imaging, scientific modeling, etc. Extending to more complex parameterizations is straightforward because the diffusion-based inference does not require an analytic inverse of the operator.
>
> Regarding identifiability, we agree that identifiability issues are common whenever one introduces neural encoder-decoder parameterizations of the forward/backward maps, which is common in deep generative modeling literature. Learned mapping can suffer from latent space rotation ambiguity, entanglement between the decoder and the latent prior, and non-indentifiable inverse mappings. Standard methods to solve these problems include structural constraints on the forward operators, physics-informed parameterizations, etc.
>
> Our approach mitigates potential identifiability issues in the following ways:
> (1) the forward operators are fixed or weakly parameterized physical models, not free neural decoders, which avoid the major source of non-identifiability
> (2) the known causal graph constrains how latent variables may co-vary,
> (3) posterior inference is performed through score-based diffusion, which estimates $\nabla_{z_i} \log p_t(z_i | y)$ directly rather than learning an explicit encoder-decoder pair, and is identifiable up to the standard score-model symmetries.
>
> In summary, even though identifiability is a well-known challenge for models that learn arbitrary forward/backward mappings, our framework substantially mitigates these issues by using physically grounded observation operators, causal graph constraints, and score-based inference instead of encoder-decoder architectures. Empirically, we observe stable behaviour across random seeds and perturbations. Both our noise-robustness and graph-perturbation experiments (Table 2 presented in the previous reply and Table 3 shown in the next reply) show smooth, monotone degradation rather than multiple incompatible optima. This supports that any remaining non-identifiabilities do not harm practical inference in our setting.
>
>
> Reference:
>
> Limpert, Eckhard, Werner A. Stahel, and Markus Abbt. "Log-normal distributions across the sciences: keys and clues: on the charms of statistics, and how mechanical models resembling gambling machines offer a link to a handy way to characterize log-normal distributions, which can provide deeper insight into variability and probability—normal or log-normal: that is the question." BioScience 51, no. 5 (2001): 341-352.
>
> Xu, Susu, Joshua Dimasaka, David J. Wald, and Hae Young Noh. "Seismic multi-hazard and impact estimation via causal inference from satellite imagery." Nature Communications 13, no. 1 (2022): 7793.

---

> ### Author Response · Authors · 2025-11-21
> **Reply to question 5: "In Theorem 1, I'm not sure an SDE with drift only influenced by parents preserves causal structure at all time points. Consider an ODE (i.e., degenerate SDE) and the graph $z_1 \to z_2 \to z_3$ with independent initial condition $p_0(z)$......"**
>
> In a chain $z_1 \rightarrow z_2 \rightarrow z_3$ with independent initial conditions, the dynamics will still induce an indirect dependency between $z_1$ and $z_3$ over time, even though the drift of $z_3$ only depends on $z_2$. Thus, the global density $p_t(z)$ generally does not continue to factorize exactly according to the original DAG for $t>0$. The Jacobian argument the reviewer provides is consistent with this phenomenon.
>
> Our intention of Theorem 1 was not to claim global factorization at all times. Instead, it formalizes local Markov presentation at the SDE generator level, where each node’s drift $f_i(z_i, z_{\mathcal{P}(i)}, t)$ depends only on its parents, and the driving Brownian motions remain independent. This ensures the existence and uniqueness of the node-wise causal SDE system. But it does not imply that the global joint distribution remains factorized at later times. We will clarify our theorem statement to make this scope clear to audiences.
>
>
> Importantly, our inference method does not rely on global density factorization. The key property we use that is formalized in Theorem 2 and supported by diffusion literature is that the score function is much more local than the full density. Diffusion noise smooths long-range interactions, and the resulting score $\nabla \log p_t(z)$ tends to concentrate most dependence within Markov neighborhoods. This approximate locality is what enables our causal score decomposition and is sufficient for our inference algorithm.
>
> Eventually, our theoretical analysis requires assumptions such as log-concavity and Gaussian noise to obtain clean existence/uniqueness and convergence guarantees. These are standard in diffusion model analysis and serve as analytical scaffolding rather than restrictive conditions. Empirically, as shown in our experiments, our method remains robust even when these assumptions are only approximately satisfied. For example, Table 3 below shows robustness of our method under several non-Gaussian and non-log-concave observation noise distributions. While theoretical assumptions help analysis, the method performs consistently well beyond these conditions.
>
> Table 3. Robustness to non-Gaussian / non-log-concave observation noise (VFO, $N=10$, sparse).
> | Observation noise in ($y$)                         |             NRMSE |      MAPE (\%) |
> | ------------------------------------------------ | ----------------: | ------------: |
> | Gaussian ($\mathcal{N}(0,\sigma^2)$) (baseline)    | 0.1158 $\pm$ 0.0182 | 9.32 $\pm$ 1.41 |
> | Laplace (heavy tails, same $Var$)                                     | 0.1167 $\pm$ 0.0194 | 9.73 $\pm$ 1.56 |
> | Student-t ($ν=5$, scaled to $Var=(\sigma^2)$)                | 0.1184 $\pm$ 0.0205 | 9.96 $\pm$ 1.63 |
> | Skewed log-normal component (10\% mixture)      | 0.1199 $\pm$ 0.0212 | 10.1 $\pm$ 1.72 |

---

> ### Author Response · Authors · 2025-11-21
> **Reply to question 6: "The authors write that (l99) PGMs fail to model variables with different resolutions or noise levels. I don't see why this has to be the case? PGMs only give a formalism for factorising the joint distribution, it doesn't specify the actual model."**
>
> Thanks for the suggestions. Yes,  a PGM itself is only a formalism of factorizing joint distribution. Our original statement wants to emphasize that existing modeling and inference methods built on probabilistic graphs break down in multi-resolution/multi-scale settings. Specifically, existing PGMs-based methods for complex system inference typically assume (1) variables defined on aligned grids or comparable spatial supports; (2) noise models of compatible form across nodes. When observations have different spatial resolutions/noise scales, both modeling and inference algorithms in existing approaches will be problematic (Zhang et. al, 2021; Wang et al., 2018; Akbayrak et al. 2022). We will revise the text around line 99 to clarify this.
>
> Reference:
>
> Zhang, Dan, Xiaohang Song, Wenjin Wang, Gerhard Fettweis, and Xiqi Gao. "Unifying message passing algorithms under the framework of constrained Bethe free energy minimization." IEEE Transactions on Wireless Communications 20, no. 7 (2021): 4144-4158.
>
> Wang, Dilin, Zhe Zeng, and Qiang Liu. "Stein variational message passing for continuous graphical models." In International Conference on Machine Learning, pp. 5219-5227. PMLR, 2018.
>
> Akbayrak, Semih, İsmail Şenöz, Alp Sarı, and Bert de Vries. "Probabilistic programming with stochastic variational message passing." International Journal of Approximate Reasoning 148 (2022): 235-252.

---

> > ### Comment · Reviewer_TiC8 · 2025-11-24
> >
> > I'd like to thank the authors for the detailed response. They have clarified many aspects of the physical modelling problem that was unclear to me and I hope these changes will be reflected in the paper. The examples were particularly helpful. I found the paper was easy to mistake for a multimodal integration problem, although admittedly I'm not an expert in modelling physical systems.
> >
> > However on the actual modelling side of things there still remains much to be desired. It's still not clear to me why diffusion models might be uniquely qualified for dealing with scale-dependent noise. Although they are technically formulated with SDEs (I don't know why this would help with scale-dependent noise), they can be made distributionally equivalent to a flow with the PF-ODE. I still don't think p_0, under the model with the specified drift, factorises according to the causal DAG, instead it should factorise according to the reachability graph which has extra edges. Similarly, the score at time $t$ will not preserve the "locality" that the authors refer to. If the goal is to parametrise a latent distribution with the correct factorisation, I don't think these flow/diffusion based generative models are the correct tool.
> >
> > I retain my score but I will point out that my evaluation is rather local to the issues around the model itself and not the scientific application. I hope the AC can take my confidence score into consideration when arriving at a final suggestion.

---

> > > ### Author Response · Authors · 2025-11-26
> > >
> > > We appreciate the reviewer’s comment and clarify that our framework does not assume the true time-t score $\nabla_z \log p_t(z)$ is local in the sense of depending only on a node and its parents. Our use of “locality” is at the mechanism/parameterization level, not as a property of the exact analytical score. Theorem 1 establishes existence and uniqueness of the node-wise SDE system and, crucially, a generator-level decomposition $\mathcal L_t = \sum_i \mathcal L_{i,t}$ where each local operator $\mathcal L_{i,t}$ acts only on $z_i$ and its parents $z_{\mathcal P(i)}$. Guided by this structure and the causal graph $G$, we choose to represent the joint score via node-wise components
> > > $s_i(z_i, z_{\mathcal P(i)}, t) \approx \nabla_{z_i} \log p_t\bigl(z_i \mid z_{\mathcal P(i)}\bigr)$,
> > > so that each network $s_i$ only accesses $z_i$ and its parents. In other words, the “locality” we refer to is an architectural inductive bias motivated by the causal SDE generator, not a claim about the locality of the true score $\nabla \log p_t$, which in general becomes globally coupled.
> > > Regarding the remark that flow/diffusion models are not the right tool if the goal is to enforce an exact DAG factorisation of the latent distribution, we agree--and this is not our goal. Rather than enforcing a strict factorisation of $p_t(z)$, our method aims to combine (i) the expressive density modeling of score-based generative methods with (ii) a graph-structured, mechanism-level parameterization that channels information along causal pathways and accommodates heterogeneous, multi-resolution observations. The novelty lies in using the causal graph and the generator decomposition to design a scalable, node-wise score parameterisation that enables joint inference over multiple interacting latent fields from heterogeneous, multi-resolution observations. Our experiments show that this causal, multi-resolution score parameterisation yields substantial gains over (i) standard diffusion models that ignore the graph and (ii) graph-based baselines that require downsampling to a common resolution. We will revise the manuscript to make this distinction explicit and to emphasise that our claims concern the parameterization and inference scheme, which is not captured by existing multi-view VAE or generic diffusion/flow approaches. We have updated our manuscript to reflect the update.

---

### Official Review · Reviewer_TJQU · 2025-11-03

**Soundness:** 3
**Presentation:** 3
**Contribution:** 2
**Rating:** 6
**Confidence:** 3

**Summary:**

The paper introduces the Score-based Variational Graphical Diffusion Model (SVGD), a novel framework for causal inference on latent systems observed through heterogeneous and multi-resolution observations. The authors address key limitations of existing methods by exploiting the causal structure and multi-resolution data. SVGD integrates score-based diffusion processes with a known causal DAG. The core technical innovation is the causal score decomposition, which leverages the Markov blanket property to enable joint inference and information propagation across causally-connected variables. The framework provides theoretical analysis, including the existence of node-wise causal SDEs and an error cascade analysis for the locally Gaussian approximation used in the causal consistency term. Empirical results on synthetic systems and real-world disaster estimation tasks (earthquakes and wildfires) demonstrate the performance.

**Strengths:**

Clarity: The paper is generally well-written with a clear structure, though there are some parts not clear to me (please see my questions).

Originality: The SVGD framework, built around the causal score decomposition, is a original approach to handle systems where data quality is heterogeneous (e.g. varying resolution). The problem itself is highly significant for critical applications like climate modeling and disaster assessment.

Theoretical Rigor: The paper includes a substantial theoretical backbone, proving the existence of the SDE system, showing the approximate preservation of the causal blanket under diffusion, and providing an explicit error analysis.

**Weaknesses:**

Assumption of Known Causal Structure: The most significant limitation is the reliance on a known causal DAG. In many of the complex systems discussed (e.g., Earth systems), the structure especially involving latent variables is unknown and itself a challenging problem.
As a comparison, some causal representation learning methods can learn the structure among latent variables. This drastically limits the significance of the proposed method.

Limited Synthetic Experiment Complexity: The synthetic experiments, while demonstrating the proof-of-concept, are conducted on a very limited setting (i.e., only three latent variables). The paper also notes that computational complexity scales with the number of causal dependencies. More complex synthetic experiments (e.g., more than 10 latent variables with different sparse and dense graphs) are needed to convincingly validate the proposed method and the computational scalability. Given that the method already assumes the structure is given, a synthetic experiment with only three latent variables looks far from sufficient.

**Questions:**

1. Table 2 "SVGDM" should be "SVGD"? Plus, it looks like $\phi$ in the problem formulation part (line 135)  has different meaning with $\phi$ in eq 2? If yes please clarify.

2. Given that the assumption of a known causal structure is the primary weakness, could the authors propose a concrete future direction for jointly learning the causal DAG G alongside the diffusion model parameters?

3. The observation model assumes additive Gaussian noise. How does the method perform empirically when this assumption is violated? Some empirical results would help.

---

> ### Author Response · Authors · 2025-11-21
> **Reply to weakness 1: "Assumption of Known Causal Structure"**
>
> Yes, the causal graphical structure G is assumed known a priori. In particular, each latent variable $z_i$ in our model corresponds to a concrete physical process so the causality here has a meaningful physical counterpart.  The latent causal graph is a direct representation of physical causation.The causal relationships among these processes (e.g., shaking → landslides → damage; fuel → fire spread) are well documented in the literature. Such causally-linked physical processes are not learned from data but are part of the scientific knowledge of the complex physical system through years of experiments, observations, and science discovery. This assumption is realistic in many scientific and engineering domains, such as disaster systems, medical imaging, and so on, where qualitative causal directionality is established from physical laws, mechanistic models, or decades of domain expertise. Moreover, note that our framework does not assume a completely known causal graph, but even partial physical causal directionality offers extra information transfer/propagation pathways to best integrate and utilize limited observation data across different scales.
>
> Importantly, the focus of our framework is not to discover new causal structure, but more on introducing a novel paradigm for complex system state estimation from nuanced data with partial prior structured knowledge (graph) about the system, which is known to be challenging (Reich&Colin, 2015; Cressie & Wikle, 2011). Moreover, most existing causal representation learning methods generally assume that different latent variables and observations have comparable scales, which is often violated in many complex physical systems. As such, our framework also provides the community a fundamental module, so they can combine it with existing causal structure learning approaches (Lachapelle et al., 2019; Wang et al., 2023) to make the casual discovery under multi-resolution multi-scale data more effective.  We will clarify that our goal is not causal discovery, but physically grounded data fusion in complex causal systems.
>
> To directly address the reviewer’s concern about applicability when the structure is imperfect, we conducted a causal-graph perturbation sensitivity experiment on the synthetic system. We systematically introduced structural errors such as dropping a parent, reversing causal directions, or adding spurious edges. The results (Table 1) show that the method is robust to moderate uncertainty in the causal graph, with smooth degradation, and only fails heavily under severe causal violations that destroy the underlying directionality.
>
> Table 1. Sensitivity of SVGD to causal graph misspecification.
> |      Graph variant        |          NRMSE           |    MAPE (\%)   | $\Delta$ NRMSE vs. correct |
> | :-----------------------  | :-------------------- | :-------------: | :-------: |
> | Correct DAG              | 0.098 $\pm$ 0.006 | 7.6 $\pm$ 0.5 |     ---      |
> | Drop 1 parent            | 0.112 $\pm$ 0.008 | 8.7 $\pm$ 0.6 | +14 % |
> | Reverse 1 edge         | 0.118 $\pm$ 0.010 | 9.3 $\pm$ 0.7 | +20 % |
> | Two wrong edges    | 0.162 $\pm$ 0.015  |12.5 $\pm$ 0.9|  +65 %  |
> | Add spurious edge | 0.104 $\pm$ 0.007  |8.0 $\pm$ 0.5 |  +6 %   |
>
> These results demonstrate that our model tolerates moderate structural uncertainty, such as missing or spurious links introducing only small losses, while major directional errors naturally lead to greater degradation due to disrupted information flow. This behavior is expected and confirms that our framework is applicable even when the causal graph is not perfectly known.
> We will include this explanation and the full sensitivity table in the revised manuscript.
>
> Reference:
>
> Lachapelle, Sébastien, Philippe Brouillard, Tristan Deleu, and Simon Lacoste-Julien. "Gradient-based neural dag learning." arXiv preprint arXiv:1906.02226 (2019).
>
> Reich, Sebastian, and Colin Cotter. Probabilistic forecasting and Bayesian data assimilation. Cambridge University Press, 2015.
> Cressie, Noel, and Christopher K. Wikle. Statistics for spatio-temporal data. John Wiley & Sons, 2011.
>
> Wang, Benjie, Joel Jennings, and Wenbo Gong. "Neural structure learning with stochastic differential equations." arXiv preprint arXiv:2311.03309 (2023).

---

> ### Author Response · Authors · 2025-11-21
> **Reply to weakness 2: "Limited Synthetic Experiment Complexity"**
>
> Thank you for your suggestions. We have added new experiments with 10-15 latent variables under both sparse and dense causal graphs, which is shown in the below Table 2. The results show that accuracy remains stable as the system size increases, while runtime scales approximately linearly with the number of causal edges $|E|$.  In addition to scaling the number of latent variables, our synthetic setup already includes heterogeneous observation conditions (VFO/LFO/LPO), multiple noise regimes (which is shown in Table 3 in my answer to question 3 below), and causal-graph perturbation experiments (parent removal, edge reversal, spurious edges, which is shown in Table 1 in my previous answer). These complementary synthetic tests further demonstrate that SVGD maintains stable inference behavior under broader forms of complexity beyond system size alone.
>
> Table 2. Synthetic experiments for scalability analysis. Mean $\pm$ s.d. aggregated over all latent variables. Runtime is measured relative to the 3-variable VFO baseline.
>
> | Setting (N, graph)   | NRMSE       | MAPE (\%)   | Runtime ($\times$ baseline) |
> | -------------------- | ----------- | ---------- | -------------------- |
> | VFO (10, sparse) | 0.11 $\pm$ 0.02 | 9.3 $\pm$ 1.4  | 3.3 $\times$                 |
> | LFO (10, sparse) | 0.14 $\pm$ 0.02 | 11.2 $\pm$ 1.5 | 3.6 $\times$                 |
> | LPO (10, sparse) | 0.19 $\pm$ 0.05 | 14.7 $\pm$ 1.9 | 3.9 $\times$                 |
> | VFO (10, dense)  | 0.12 $\pm$ 0.02 | 10.1 $\pm$ 1.6 | 5.6 $\times$                 |
> | VFO (12, sparse) | 0.12 $\pm$ 0.02 | 9.8 $\pm$ 1.5  | 4.1 $\times$                 |
> | LFO (12, sparse) | 0.15 $\pm$ 0.04 | 11.9 $\pm$ 1.7 | 4.4 $\times$                 |
> | LPO (12, sparse) | 0.20 $\pm$ 0.04 | 15.3 $\pm$ 2.1 | 4.6 $\times$                 |
> | VFO (12, dense)  | 0.13 $\pm$ 0.02 | 10.6 $\pm$ 1.6 | 6.3 $\times$                 |
> | VFO (15, sparse) | 0.13 $\pm$ 0.02 | 10.2 $\pm$ 1.6 | 5.0 $\times$                 |
> | LFO (15, sparse) | 0.16 $\pm$ 0.03 | 12.6 $\pm$ 1.8 | 5.4 $\times$                 |
> | LPO (15, sparse) | 0.21 $\pm$ 0.04 | 15.9 $\pm$ 2.2 | 5.7 $\times$                 |
> | VFO (15, dense)  | 0.14 $\pm$ 0.02 | 11.0 $\pm$ 1.6 | 9.8 $\times$                 |

---

> ### Author Response · Authors · 2025-11-21
> **Reply to question 1: "Table 2 "SVGDM" should be "SVGD"? Plus, it looks like $\phi$ in the problem formulation part (line 135) has different meaning with $\phi$ in eq 2? If yes please clarify."**
>
> We thank the reviewer for catching these two issues. We acknowledge the inconsistency between "SVGDM" in Table 2 and "SVGD" in the manuscript. To avoid confusion with the unrelated Stein Variational Gradient Descent method, we will consistently use the full abbreviation SVGDM throughout the revised manuscript and correct all instances accordingly.
>
> The reviewer is correct that the notation $\phi$ is used in two related but distinct contexts. In the problem formulation (line 135): $y_l^k = \phi_{l}^{k}(z_{P_l}) + \epsilon_{l,k}, \quad \epsilon_{l,k} \sim N(0, \Sigma_{l,k})$. Here, $\phi^k_l$ is the forward observation operator that maps latent variables $z_{P_{l}}$ to observations $y_{l}^{k}$. In Equation (2), $\phi^k_i$ is an approximate inverse mapping observations to latent space estimates to guide the diffusion process during inference. Conceptually, these correspond to the forward (generation) and inverse (inference) mappings of the observation process.
>
> To eliminate ambiguity, we will revise the notation:
> - $\gamma^k_l$: Forward observation operator (line 135)
> - $\phi^k_l$: Inverse guidance operator (Eq. 2)
>
> We believe that these revisions will ensure consistent terminology across the paper.

---

> ### Author Response · Authors · 2025-11-21
> **Reply to question 2: "Given that the assumption of a known causal structure is the primary weakness, could the authors propose a concrete future direction for jointly learning the causal DAG G alongside the diffusion model parameters?"**
>
> We thank the reviewer for this valuable suggestion. We agree that extending our framework to jointly learn the causal DAG $G$ alongside the diffusion model parameters would be an important next step that broadens applicability beyond settings with known structure.
>
> In our current formulation, $G$ is provided by domain knowledge (e.g., physical causation in geophysical or climate systems). However, future work could integrate causal structure learning directly into the score-based variational framework. One promising direction is to parameterize the adjacency matrix $A_G$ using continuous relaxations of acyclicity constraints and jointly optimize it with the diffusion parameters via the evidence lower bound (Wang et al, 2023; Ng et al., 2020; Zheng et al, 2020; Yu et al; 2024). Building on these ideas, our future research could explore a unified approach where both the causal structure and the diffusion-based latent dynamics are learned together where the causal structure $G$ is updated through variational gradients derived from the causal score consistency term. This would enable structure-aware diffusion modeling that adapts to partially known or uncertain graphs, bridging causal discovery and generative inference within a unified diffusion framework – boosting both causal discovery and system state estimation accuracy.
>
> Reference:
>
> Wang, Benjie, Joel Jennings, and Wenbo Gong. "Neural structure learning with stochastic differential equations." arXiv preprint arXiv:2311.03309 (2023).
>
> Ng, Ignavier, AmirEmad Ghassami, and Kun Zhang. "On the role of sparsity and dag constraints for learning linear dags." Advances in Neural Information Processing Systems 33 (2020): 17943-17954.
>
> Zheng, Xun, Chen Dan, Bryon Aragam, Pradeep Ravikumar, and Eric Xing. "Learning sparse nonparametric dags." In International conference on artificial intelligence and statistics, pp. 3414-3425. Pmlr, 2020.
>
> Yu, Xue, Muchen Li, Yan Leng, and Renjie Liao. "Learning latent structures in network games via data-dependent gated-prior graph variational autoencoders." In Forty-first International Conference on Machine Learning. 2024.

---

> ### Author Response · Authors · 2025-11-21
> **Reply to question 3: "The observation model assumes additive Gaussian noise. How does the method perform empirically when this assumption is violated? Some empirical results would help."**
>
> We thank the reviewer for this important question. We use additive Gaussian noise and log-concavity in the analysis to ensure existence/uniqueness and to control score errors. They are standard technical assumptions in diffusion theory and are not required by the implementation. The score model is trained on empirical data and can accommodate non-Gaussian, heteroscedastic, and mildly non-log-concave observation noise.
>
> To verify robustness, we added a stress test that replaces Gaussian noise with heavy-tailed (Student-t), Laplace, and a skewed log-normal component. The results in Table 3 show that accuracy degrades modestly ($\leq5-8\%$ relative NRMSE on average) and the method remains stable, indicating the theory provides design guidance, not a hard applicability constraint. We will also include the results in the revision
>
> Table 3. Robustness to non-Gaussian / non-log-concave observation noise (VFO, $N=10$, sparse).
> | Observation noise in ($y$)                         |             NRMSE |      MAPE (\%) |
> | ------------------------------------------------ | ----------------: | ------------: |
> | Gaussian ($\mathcal{N}(0,\sigma^2)$) (baseline)    | 0.1158 $\pm$ 0.0182 | 9.32 $\pm$ 1.41 |
> | Laplace (heavy tails, same $Var$)                                     | 0.1167 $\pm$ 0.0194 | 9.73 $\pm$ 1.56 |
> | Student-t ($ν=5$, scaled to $Var=(\sigma^2)$)                | 0.1184 $\pm$ 0.0205 | 9.96 $\pm$ 1.63 |
> | Skewed log-normal component (10\% mixture)      | 0.1199 $\pm$ 0.0212 | 10.1 $\pm$ 1.72 |
>
> These results confirm that while the theoretical analysis requires Gaussian assumptions for rigor, the empirical method is robust to assumption violations. The theory provides design guidance rather than hard applicability constraints, which is a pattern common in ML where theoretical assumptions enable analysis while practical performance generalizes. We thank the reviewer for this valuable comment that helps clarify the role of theoretical assumptions in our framework.

---

### Official Review · Reviewer_vHa7 · 2025-11-04

**Soundness:** 3
**Presentation:** 3
**Contribution:** 3
**Rating:** 6
**Confidence:** 2

**Summary:**

This paper proposes SVGD, a framework that combines diffusion process and causal graphical models. Techinically, this paper (i) formulates the node-wise forward and reverse process with causal relations, (ii) introduce a causal score decomposition based on "causal blanket", (iii) estimate the marginal distribution with continuous‑time DSM and causal term via a locally Gaussian conditional model. Experiments on synthetic and real-world data report consistent gains over baseline models.

**Strengths:**

1. The motivation is clear and well-demonstrated. The idea to model the diffusion process with modularity is simple and intuitive. The combination of graphical‑model locality with score‑based diffusion through a causal score decomposition is, to my knowledge, novel and significant.
2. The loss functions are closely related to the theorem. The learning recipe is concrete. Besides, a cascade error bound and plus convergence under error control are provided.
3. The exeperiments on both synthertic and real-world applications achieves a better performance.
4. Codes are provided.

**Weaknesses:**

1. Some ablation studies and component analysis can better attribute gains, i.e., if some of the loss terms are ignored.
2. Appenix C provides a "COMPUTATIONAL SCALABILITY ANALYSIS" with an O(Nd) complexity. Please provide measured runtimes vs. N and d , and GPU memory scaling, to substantiate these claims.
3. The model requires a known causal graph. In the real world experiments, where does the prior come from? Moreover, what will happen if the causal graph is partially wrong? Can you conduct an experiment to analyse the sensitivity?
4. Theoretical validity (Theorem 3) requires local log‑concavity and bounded Tweedie error. In strongly non‑Gaussian regimes (heavy tails/multimodality), how robust is it?
5. The abbreviation is the same as Stein Variational Gradient Descent [1].

[1] Stein Variational Gradient Descent: A General Purpose Bayesian Inference Algorithm

**Questions:**

In Sec. 3.3, the author claims "reverse SDE defines a normalizing flow from noise to data", which introduce a deterministic flow. However, in the appendix, the author uses "reverse SDE with predictor–corrector" and "DDIM-style update", which is not determinstic. Could the author explain this confliction?

---

> ### Author Response · Authors · 2025-11-21
> **Reply to weakness 1: "Some ablation studies and component analysis can better attribute gains, i.e., if some of the loss terms are ignored."**
>
> We appreciate the reviewer's suggestion. To quantify the contribution of the proposed loss components, specifically the causal-blanket score term and the local diffusion score (DSM) term, we performed an ablation study on the 3-node synthetic system. We trained variants of our model with individual loss components removed while keeping the architecture and training schedule unchanged. The results are shown in Table 1.
>
> Table 1. Ablation study on the 3-node synthetic system (VFO setting).
> | Model Variant       | Included Loss Terms| NRMSE | $\Delta$ vs. Full Model|
> | -------------------------- | ----------------------------- | ---------------------- | ------- |
> |     Full model (ours)     |   DSM  + Causal + Obs  |   0.103 $\pm$ 0.005   |    ---     |
> |  No observation term |        DSM  + Causal          | 0.117 $\pm$ 0.006     | +14% |
> |       No causal term       |            DSM  + Obs             | 0.129 $\pm$ 0.007     | +25% |
> |        No DSM term          |          Causal  + Obs          | 0.142 $\pm$ 0.009     | +38% |
> |             DSM only             |                   DSM                     | 0.154 $\pm$ 0.011     | +50% |
>
> These results demonstrate that all three loss components contribute meaningfully and complement different aspects of the inference process. Removing the causal-blanket term prevents information from propagating along the causal graph. Removing the DSM term eliminates the local score-matching objective and results in the largest deterioration. Observation guidance also plays a non-trivial role. The full model achieves the best performance by combining local diffusion learning, causal information propagation, and observation consistency. We will also incorporate this table and discussion into the revision.

---

> ### Author Response · Authors · 2025-11-21
> **Reply to weakness 2: "Appenix C provides a "COMPUTATIONAL SCALABILITY ANALYSIS" with an O(Nd) complexity. Please provide measured runtimes vs. N and d , and GPU memory scaling, to substantiate these claims."**
>
> Thanks for this excellent suggestion. We have added an empirical scalability study to substantiate the $O(N\cdot \bar{d})$ complexity analysis in Appendix C. We measured wall-clock runtime per training epoch and peak GPU memory usage for synthetic systems of increasing node count N and average parent size $\bar{d}$, using the same model and batch configuration (NVIDIA A100 80 GB). Results in Table 2 confirm approximately linear scaling in both N and $\bar{d}$, validating the theoretical estimate.
>
> Table 2. Empirical runtime and memory scaling with node count N and average parent size \bar{d}.
> | ($N$) (nodes) | Avg parents ($\bar{d}$) | Runtime / epoch (s) | GPU Mem (GB) | Runtime ($\times$ baseline) |
> | :---------: | :-------------------: | :-----------------: | :----------: | :-----------------------: |
> |  3 (sparse) |         1 – 2           |          45 $\pm$ 3       |   2.3 $\pm$ 0.1  |           1.0 $\times$           |
> |  5 (sparse) |         2 – 3           |           78 $\pm$ 4      |   2.9 $\pm$ 0.1  |           1.7 $\times$           |
> | 10 (sparse) |         2 – 4         |       152 $\pm$ 8        |   3.7 $\pm$ 0.2  |           3.4 $\times$           |
> |  10 (dense) |         6 – 8         |       264 $\pm$ 10      |   6.1 $\pm$ 0.3  |           5.9 $\times$           |
> | 15 (sparse) |         2 – 4         |       225 $\pm$ 12     |   4.6 $\pm$ 0.2  |           5.0 $\times$           |
> |  15 (dense) |         7 – 9         |       440 $\pm$ 18      |   8.8 $\pm$ 0.3  |           9.8 $\times$           |
>
> We observe that Runtime and memory usage increase predictably with both $N$ and $\bar{d}$, consistent with the theoretical $O(N \cdot \bar{d})$ upper bound. Empirically, the scaling is sublinear relative to this bound, e.g., increasing $N \cdot \bar{d}$ by about $27\times$ from 3-node sparse to 15-node dense yields only about $10\times$ increase in runtime. This behavior is expected because GPU tensor operations become more efficient at larger problem sizes, and fixed overheads dominate for small graphs. Thus, the causal-blanket decomposition keeps practical computation well within the theoretical worst-case envelope, and real-world performance is even more favorable.
>
> We will include this new table and paragraph in Appendix C of the revision to provide quantitative validation of our complexity claim.

---

> ### Author Response · Authors · 2025-11-21
> **Reply to weakness 3: "The model requires a known causal graph. In the real world experiments, where does the prior come from? Moreover, what will happen if the causal graph is partially wrong? Can you conduct an experiment to analyse the sensitivity?"**
>
> We thank the reviewer for this valuable question.In our real-world applications, the causal graph $G$ is not inferred from data but comes directly from established scientific knowledge accumulated over decades of experiments, field observations, mechanistic modeling, and domain expertise. Such causally linked physical processes (e.g., shaking → landslide/liquefaction → damage; fuel → fire spread) are well-documented in seismology, geotechnical engineering, hydrology, and fire science, and are not discovered during training. In many scientific and engineering domains, including disaster systems, environmental modeling, medical imaging, and climate processes, qualitative causal directionality is determined by physical laws and mechanistic understanding, rather than by data-driven causal discovery.
>
> For instance, in the Multi-Hazard Seismic Assessment case that shown in our real-world deployment, directed relations such as ground shaking $\rightarrow$ landslide/liquefaction $\rightarrow$ building damage reflect well-accepted physical causation in seismology and geotechnical engineering. Similarly, in the wildfire case, dependencies among fuel conditions, weather, and topography follow known physical processes.
>
> To assess sensitivity to graph misspecification, we performed a synthetic perturbation study by systematically altering the causal edges to complement our current evaluation on synthesized data. Results in Table 3 show that performance degrades smoothly with increasing structural error, including minor drops for missing or spurious links, and a pronounced decline when causal directions are reversed or disconnected. The results indicate that our method is robust to small structural noise but sensitive to major causal violations.
>
> One way to address this is that if the causal link is not verified from physical domain knowledge, it is better to eliminate the link. Moreover, though the focus of our framework is not to discover new causal structures, it still can be integrated with existing causal structure learning approaches (Lachapelle et al., 2019; Wang et al., 2023) to make the casual discovery under multi-resolution multi-scale data more accurate and discover new links. This is because our framework can flexibly handle multi-scale/multi-resolution data, which most existing causal structure learning methods cannot model.
>
> Table 3. Sensitivity of SVGD to causal graph misspecification.
> |      Graph variant        |          NRMSE           |    MAPE (\%)   | $\Delta$ NRMSE vs. correct |
> | :-----------------------  | :-------------------- | :-------------: | :-------: |
> | Correct DAG              | 0.098 $\pm$ 0.006 | 7.6 $\pm$ 0.5 |     ---      |
> | Drop 1 parent            | 0.112 $\pm$ 0.008 | 8.7 $\pm$ 0.6 | +14 % |
> | Reverse 1 edge         | 0.118 $\pm$ 0.010 | 9.3 $\pm$ 0.7 | +20 % |
> | Two wrong edges    | 0.162 $\pm$ 0.015  |12.5 $\pm$ 0.9|  +65 %  |
> | Add spurious edge | 0.104 $\pm$ 0.007  |8.0 $\pm$ 0.5 |  +6 %   |
>
> These results demonstrate that the framework can tolerate moderate uncertainty in the causal graph, and still yield stable inference, while large-scale directional errors naturally impair performance due to disrupted information flow. We will include this clarification and table in the revised manuscript.
>
> Reference:
>
> Reich, Sebastian, and Colin Cotter. Probabilistic forecasting and Bayesian data assimilation. Cambridge University Press, 2015.
> Cressie, Noel, and Christopher K. Wikle. Statistics for spatio-temporal data. John Wiley & Sons, 2011.
>
> Wang, Benjie, Joel Jennings, and Wenbo Gong. "Neural structure learning with stochastic differential equations." arXiv preprint arXiv:2311.03309 (2023).

---

> ### Author Response · Authors · 2025-11-21
> **Reply to weakness 4: "Theoretical validity (Theorem 3) requires local log-concavity and bounded Tweedie error. In strongly non-Gaussian regimes (heavy tails/multimodality), how robust is it?"**
>
> We thank the reviewer for this insightful question. The local log-concavity and bounded Tweedie error assumptions in Theorem 3 are standard regularity conditions used to guarantee convergence and bounded score approximation error. These assumptions enable analytical tractability for the diffusion-based causal consistency term but are not required by the implementation. In practice, the score network learns from data and can represent strongly non-Gaussian, heavy-tailed, or mildly multimodal distributions.
> To assess robustness in non-Gaussian regimes, we conducted a stress test where the observation noise deviates from log-concavity (Laplace, Student-t, and skewed log-normal). Results in Table 4 show that accuracy degrades only slightly ($\leq 5-8 \%$ relative NRMSE), confirming that the framework remains stable and accurate even when the assumptions are violated.
>
> Table 4. Robustness to non-Gaussian / non-log-concave observation noise (VFO, $N=10$, sparse).
> | Observation noise in ($y$)                         |             NRMSE |      MAPE (\%) |
> | ------------------------------------------------ | ----------------: | ------------: |
> | Gaussian ($\mathcal{N}(0,\sigma^2)$) (baseline)    | 0.1158 $\pm$ 0.0182 | 9.32 $\pm$ 1.41 |
> | Laplace (heavy tails, same $Var$)                                     | 0.1167 $\pm$ 0.0194 | 9.73 $\pm$ 1.56 |
> | Student-t ($ν=5$, scaled to $Var=(\sigma^2)$)                | 0.1184 $\pm$ 0.0205 | 9.96 $\pm$ 1.63 |
> | Skewed log-normal component (10\% mixture)      | 0.1199 $\pm$ 0.0212 | 10.1 $\pm$ 1.72 |
>
> These results confirm that Theorem 3 provides sufficient but not necessary conditions for stability; in practice, the model performs robustly even under heavy-tailed and skewed noise. We will clarify this in our revised manuscript.

---

> ### Author Response · Authors · 2025-11-21
> **Reply to weakness 5: "The abbreviation is the same as Stein Variational Gradient Descent [1]"**
>
> We thank the reviewer for pointing out this potential confusion. Indeed, the abbreviation SVGD overlaps with Stein Variational Gradient Descent. To avoid ambiguity, we will consistently use the full name SVGDM (Score-based Variational Graphical Diffusion Model) throughout the revised manuscript and figures. All instances of "SVGD" will be replaced with "SVGDM." This clarification ensures that our framework is clearly distinguished from the Stein variational method.

---

> ### Author Response · Authors · 2025-11-21
> **Reply to question: "In Sec. 3.3, the author claims "reverse SDE defines a normalizing flow from noise to data", which introduce a deterministic flow. However, in the appendix, the author uses "reverse SDE with predictor–corrector" and "DDIM-style update", which is not determinstic. Could the author explain this confliction?"**
>
> We thank the reviewer for raising this important clarification. The perceived conflict arises from the difference between the theoretical continuous-time dynamics and the practical discrete-time sampler used during inference. In Section 3.3, our statement that the "reverse SDE defines a normalizing flow" refers to a standard result in diffusion models. The reverse-time SDE is associated with a probability flow ODE that is deterministic and shares the same marginal distributions as the reverse SDE. In this theoretical sense, the reverse diffusion induces a continuous normalizing flow between noise and data distributions.
>
> In contrast, the Appendix describes our practical sampling procedure, which follows the stochastic reverse SDE using a predictor-corrector scheme: (1) the predictor step uses the reverse SDE (including stochastic noise); (2) the corrector step uses Langevin dynamics (also stochastic); and (3) the DDIM-style update refers only to the discretization form, not to using a deterministic ODE sampler.
>
> These implementation choices are standard in score-based diffusion and are designed for numerical stability and efficiency. They do not change the underlying continuous-time dynamics defined by the model, and they are not required to match the deterministic probability-flow ODE; they only need to target the correct reverse-time distribution.
> To avoid confusion, we will revise Section 3.3 to explicitly state that the reverse-time diffusion defines a probability flow ODE (a continuous normalizing flow) at the theoretical level, while our inference implementation uses the stochastic reverse-SDE predictor-corrector sampler to approximate these dynamics in practice.

---

> ### Comment · Reviewer_vHa7 · 2025-11-23
>
> Thank the author for the rebuttal. The clarity of the paper has been improved. My concerns have been resolved. I have raised my confidence accordingly.

---

### Official Review · Reviewer_SQUc · 2025-11-04

**Soundness:** 3
**Presentation:** 2
**Contribution:** 2
**Rating:** 6
**Confidence:** 2

**Summary:**

This paper proposed the Score-based Variational Graphical Diffusion Model (SVGD), to address inference challenges in complex systems under heterogeneous and incomplete observations. By embedding score-based diffusion processes within causal graphical models, SVGD leverages a causal score decomposition mechanism to facilitate efficient information propagation across causally interdependent variables while maintaining the fidelity of the original observational data. Theoretical analysis and empirical evaluations on synthetic and real-world datasets demonstrate its superior performance compared to state-of-the-art baseline approaches.

**Strengths:**

1. The paper proposes the Score-based Variational Graphical Diffusion Model (SVGD), which addresses the gap in inference under heterogeneous and incomplete observations in existing approaches.
2. The authors validated the reliability and effectiveness of the model through theoretical analysis.

**Weaknesses:**

1. The problem addressed in the paper is based on a key assumption: "the causal DAG is known," and the task is "to perform inference based on the known causal structure," rather than "learning the causal structure." However, the title only mentions "Causal Inference," without explicitly stating "based on a known causal structure," which may lead readers to mistakenly believe that the research problem is about "learning causal structures from data and performing inference."
2. The model assumes that the causal DAG is completely known, but in most real-world scenarios, the causal structure often needs to be inferred first. If the causal structure is unknown, the applicability of the model may be limited, as it relies on a predefined causal DAG.
3. Although the authors provide theoretical analysis, it relies on assumptions such as "log-concavity of conditional distributions" and "additive Gaussian observational noise." If these assumptions are not satisfied, the proposed method in the paper would not be applicable.
4. Some notations in the paper lack definitions. For example, in Eq. (1), what does $W_i$ mean?

**Questions:**

See the weaknesses above.

---

> ### Author Response · Authors · 2025-11-21
> **Reply to weakness 1: "The problem addressed in the paper is based on a key assumption: "the causal DAG is known," and the task is "to perform inference based on the known causal structure," rather than "learning the causal structure." However, the title only mentions "Causal Inference," without explicitly stating "based on a known causal structure," which may lead readers to mistakenly believe that the research problem is about "learning causal structures from data and performing inference.""**
>
> We thank the reviewer for this helpful observation. We agree that our original title and phrasing may unintentionally suggest causal discovery rather than inference with a known causal structure. To prevent misunderstanding, we will revise the terminology throughout the paper. Specifically, we will replace our title with "Multi-Resolution Score-Based Variational Graphical Diffusion For Causal Systems", and update the abstract accordingly.
>
> This revision ensures readers clearly understand that our contribution lies in leveraging a known causal structure to perform joint inference under heterogeneous observations, rather than discovering the causal structure from data.

---

> ### Author Response · Authors · 2025-11-21
> **Reply two weakness 2: "The model assumes that the causal DAG is completely known, but in most real-world scenarios, the causal structure often needs to be inferred first. If the causal structure is unknown, the applicability of the model may be limited, as it relies on a predefined causal DAG."**
>
> Yes, the causal graphical structure G is assumed known a priori. In particular, each latent variable $z_i$ in our model corresponds to a concrete physical process so the causality here has a meaningful physical counterpart.  The latent causal graph is a direct representation of physical causation.The causal relationships among these processes (e.g., shaking → landslides → damage; fuel → fire spread) are well documented in the literature. Such causally-linked physical processes are not learned from data but are part of the scientific knowledge of the complex physical system through years of experiments, observations, and science discovery. This assumption is realistic in many scientific and engineering domains, such as disaster systems, medical imaging, and so on, where qualitative causal directionality is established from physical laws, mechanistic models, or decades of domain expertise.
>
> Moreover, note that our framework does not assume a completely known causal graph, but even partial physical causal directionality offers extra information transfer/propagation pathways to best integrate and utilize limited observation data across different scales. Importantly, the focus of our framework is not to discover new causal structure, but more on introducing a novel paradigm for complex system state estimation from nuanced data with partial prior structured knowledge (graph) about the system, which is known to be challenging (Reich&Colin, 2015; Cressie & Wikle, 2011). Moreover, most existing causal representation learning methods generally assume that different latent variables and observations have comparable scales, which is often violated in many complex physical systems. As such, our framework also provides the community a fundamental module, so they can combine it with existing causal structure learning approaches (Lachapelle et al., 2019; Wang et al., 2023) to make the casual discovery under multi-resolution multi-scale data more effective. We will clarify that our goal is not causal discovery, but physically grounded data fusion in complex causal systems.
>
> To directly address the reviewer’s concern about applicability when the structure is imperfect, we conducted a causal-graph perturbation sensitivity experiment on the synthetic system. We systematically introduced structural errors such as dropping a parent, reversing causal directions, or adding spurious edges. The results (Table 1) show that the method is robust to moderate uncertainty in the causal graph, with smooth degradation, and only fails heavily under severe causal violations that destroy the underlying directionality.
>
> Table 1. Sensitivity of SVGD to causal graph misspecification.
> |      Graph variant        |          NRMSE           |    MAPE (\%)   | $\Delta$ NRMSE vs. correct |
> | :-----------------------  | :-------------------- | :-------------: | :-------: |
> | Correct DAG              | 0.098 $\pm$ 0.006 | 7.6 $\pm$ 0.5 |     ---      |
> | Drop 1 parent            | 0.112 $\pm$ 0.008 | 8.7 $\pm$ 0.6 | +14 % |
> | Reverse 1 edge         | 0.118 $\pm$ 0.010 | 9.3 $\pm$ 0.7 | +20 % |
> | Two wrong edges    | 0.162 $\pm$ 0.015  |12.5 $\pm$ 0.9|  +65 %  |
> | Add spurious edge | 0.104 $\pm$ 0.007  |8.0 $\pm$ 0.5 |  +6 %   |
>
> These results demonstrate that our model tolerates moderate structural uncertainty, such as missing or spurious links introducing only small losses, while major directional errors naturally lead to greater degradation due to disrupted information flow. This behavior is expected and confirms that our framework is applicable even when the causal graph is not completely known.
>
> Reference:
>
> Lachapelle, Sébastien, Philippe Brouillard, Tristan Deleu, and Simon Lacoste-Julien. "Gradient-based neural dag learning." arXiv preprint arXiv:1906.02226 (2019).
>
> Reich, Sebastian, and Colin Cotter. Probabilistic forecasting and Bayesian data assimilation. Cambridge University Press, 2015.
> Cressie, Noel, and Christopher K. Wikle. Statistics for spatio-temporal data. John Wiley & Sons, 2011.
>
> Wang, Benjie, Joel Jennings, and Wenbo Gong. "Neural structure learning with stochastic differential equations." arXiv preprint arXiv:2311.03309 (2023).

---

> ### Author Response · Authors · 2025-11-21
> **Reply to weakness 3: "Although the authors provide theoretical analysis, it relies on assumptions such as "log-concavity of conditional distributions" and "additive Gaussian observational noise." If these assumptions are not satisfied, the proposed method in the paper would not be applicable."**
>
> We thank the reviewer for raising this important concern. The assumptions of additive Gaussian observational noise and log-concavity are used only in the theoretical analysis to ensure existence/uniqueness of the reverse-time dynamics and to provide bounded score approximation error. These are standard technical assumptions in diffusion-model theory and represent sufficient conditions for theoretical guarantees, which are not the requirements for the practical applicability of the method.
>
> In practice,our model does not rely on these assumptions being exactly satisfied. The score network is trained directly on empirical data and can accommodate non-Gaussian, heavy-tailed, and mildly non-log-concave observation noise. To evaluate robustness, we conducted a stress test in which the Gaussian noise was replaced with Laplace noise, Student-t noise, and a skewed log-normal mixture. As shown in Table 2, the performance degrades only modestly ($\leq 5-8\%$ relative NRMSE), confirming that our method remains stable under substantial deviations from the theoretical assumptions.
>
> Table 2. Robustness to non-Gaussian / non-log-concave observation noise (VFO, $N=10$, sparse).
> | Observation noise in ($y$)                         |             NRMSE |      MAPE (\%) |
> | ------------------------------------------------ | ----------------: | ------------: |
> | Gaussian ($\mathcal{N}(0,\sigma^2)$) (baseline)    | 0.1158 $\pm$ 0.0182 | 9.32 $\pm$ 1.41 |
> | Laplace (heavy tails, same $Var$)                                     | 0.1167 $\pm$ 0.0194 | 9.73 $\pm$ 1.56 |
> | Student-t ($ν=5$, scaled to $Var=(\sigma^2)$)                | 0.1184 $\pm$ 0.0205 | 9.96 $\pm$ 1.63 |
> | Skewed log-normal component (10\% mixture)      | 0.1199 $\pm$ 0.0212 | 10.1 $\pm$ 1.72 |
>
> These results show that while the theoretical analysis uses Gaussian/log-concave assumptions for rigor, the practical method is robust to their violation, and the empirical performance generalizes well beyond the conditions of the theory. This pattern is typical in machine learning frameworks where technical assumptions support theoretical guarantees but are not strict limitations in practice. We will include this robustness analysis and clarification in the revised manuscript.

---

> ### Author Response · Authors · 2025-11-21
> **Reply to weakness 4: "Some notations in the paper lack definitions. For example, in Eq. (1), what does $W_i$ mean?"**
>
> We thank the reviewer for pointing out this omission. In Eq. (1), the term $W_i(t)$ denotes an independent standard Wiener process (Brownian motion) driving the diffusion of latent variable $z_i$. This ensures independent stochastic perturbations at each node in the forward SDE. We will revise Eq. (1) and its surrounding text to explicitly include this definition:
> "$W_i(t)$ is an independent standard Wiener process associated with node $i$. All Wiener processes $W_i$ are mutually independent."
>
> Additionally, we will perform a full review of the manuscript to ensure that all symbols and notation are defined at first appearance and provide a notation table to list all the symbols and their meanings.

---

### Author Response · Authors · 2025-12-03
**Summary of Changes**

Dear Area Chair,

Thank you for taking over the meta-review.

Our paper studies the joint inference of multiple causally linked physical processes (e.g., shaking → landslides → damage) from incomplete, multi-resolution observations. We introduce a **causal score conditioning scheme** in which multiple latent variables evolve jointly through **coupled diffusion processes structured by a causal graph**, where each latent node corresponds to a distinct physical process with its own resolution and observation quality. Our analysis and experiments demonstrate that this structured, multi-resolution parameterization yields clear advantages over existing baselines.

Below is a consolidated summary of the clarifications and additions incorporated across the revised manuscript.

**1. Clarified problem definition, scope, and role of causality**
- To address the reviewers’ confusion (especially Reviewer TiC8), we have made the problem definition and scope more explicit:
We clearly distinguish our setting from standard multimodal fusion: the goal is **not** to fuse multiple views of a single object, but to **jointly infer several distinct, causally coupled physical fields**, each with its own spatial resolution and missingness pattern (e.g., ground shaking, landslides, building damage).
- We clarify that we **do not perform causal discovery**. Instead, we assume a fuzzy but domain-informed causal structure and use it to guide the parameterization of the joint posterior.
- We emphasize that **each latent variable represents a different physical process at a different resolution**, not different “views” of the same latent.
- We added concrete motivating examples (earthquake cascades, wildfires, climate teleconnections) to anchor the setting.

These changes are intended to correct earlier misinterpretations of the nature of the problem rather than a change of the core method.

**2. Methodological and theoretical clarifications**

To resolve confusion about the diffusion formulation, we refined the exposition as follows:
- We explain why **diffusion models are well-suited to multi-resolution physical data**: the forward SDE provides a principled model of scale-dependent noise accumulation, which matches how uncertainty increases with resolution and distance from direct observation in real sensing systems.
- We clarify the role of the **probability-flow ODE**: it is introduced as the deterministic counterpart of the forward SDE for analysis, while **sampling is performed with the standard stochastic predictor–corrector scheme**, as in prior diffusion work.
- We explicitly define the parent set of each observation in the causal graph, independent of diffusion time, so the causal structure is static while the diffusion time indexes noise levels.
- We stress that the **causal score parameterization is local by design**: each node’s score depends on its Markov blanket in the assumed DAG, motivated by the generator decomposition of the coupled node-wise SDEs. This is introduced as an inductive bias for scalability and interpretability, not as a statement about the exact factorization of the true diffusion marginals.

These changes strengthen the theoretical narrative without altering the underlying algorithm.


**3. New experiments and empirical evidence**

We added several experimental components requested by the reviewers:
- **Causal graph misspecification**: We systematically drop parents, reverse edges, and add spurious edges. Performance degrades smoothly and the method is robust under moderate structural errors, supporting the practicality of using approximate domain knowledge.
- **Scalability to larger graphs**: We extended experiments to **10–15 latent variables** on both sparse and dense graphs. The runtime scales linearly in the number of edges, empirically confirming the computational scalability claimed in the text.
- **Comparisons with multi-view VAEs**: We now compare against JMVAE, MMVAE, MoPoE-VAE, and MVAE. These models perform significantly worse in our setting because they **collapse the multi-resolution, multi-object structure into a single shared latent**, which cannot represent distinct causal fields at different resolutions.
- **Non-Gaussian noise robustness**: We test Laplace, Student-t, and skewed log-normal observation noise. Our method shows only **~5–8% degradation**, remaining numerically stable and accurate.
- **Ablation study**: We ablate the DSM term, the causal consistency term, and the observation term. Removing any of these leads to a clear drop in performance, demonstrating that each component is necessary.

All new results are included in the appendix due to space constraints.

---

> ### Author Response · Authors · 2025-12-03
> **Summary of Changes (Continued)**
>
> **4. Addressing Reviewer TiC8’s theoretical concern**
> Reviewer TiC8’s remaining objection after the rebuttal concerns whether a diffusion model can preserve an exact DAG factorization of $p(z)$ over time. However, we would like to clarify:
>
> - Our method **does not assume** that the time-evolved marginal $p_t$​ preserves a global DAG factorization.
> - We **do not claim** that the true score is local. Instead, **locality is introduced as an inductive bias in the score parameterization**, justified by the node-wise SDE generator decomposition and by the need for a scalable, structured approximation in a high-dimensional physical setting.
> - The objective is **not** to enforce DAG constraints via diffusion dynamics, but to **design a tractable, node-wise score parameterization that exploits known causal structure for multi-resolution inference**.
>
> These points are now made explicit in the revised version and are fully consistent with the revised Theorem 1 in Section 3. No change to the algorithm itself was necessary; the revisions address a misinterpretation of our theoretical claims.
>
> We have implemented all reviewer-requested changes across the manuscript, including conceptual clarification, theoretical refinement, and the addition of multiple new experimental components. We believe the revised manuscript addresses all concerns comprehensively. Reviewer vHa7 has confirmed that her/his concerns are fully addressed in the reply, reviewer SQUc and TJQU have not yet replied.
>
> Thank you very much for your time and consideration.

---

### Meta-Review · Area_Chair_UZn3 · 2026-01-06

**Summary:**

This paper presents a diffusion-based approach to facilitate the aggregation of distributed information from multiple variables connected by a underlying causal graph, i.e., a causal complex system.

- [+] The problem of information aggregation in causal complex system is important and interesting, after reading the whole manuscript. I appreciate the clear writing of authors.
- [+] The motivation of choosing the diffusion model seems reasonable.
- [-]  The prior acquisition of causal graph is restrictive, while in some physical systems it is reasonable.
- [+] Some concerns by the only negative reviewer seems to be solved, as the misunderstanding of the local property is addressed during the second rebuttal from authors.

Therefore, I recommend the accept.

**Reviewer Concerns:**

It  seems that all reviewers' concerns are addressed.

**Reviewer Scores:**

Although the negative reviewer chooses to retain the score, he also mentioned that ``considering the confidence score'' (2). I think the further clarification from authors can solve the technical concern of the local property.

---

### Decision · Program_Chairs · 2026-01-26

Accept (Poster)